# Discrete Interpolants: Unifying the Masked Generative and Discrete Diffusion Models

## Abstract

In generative models, two paradigms have gained attraction in various applications: next-set prediction-based Masked Generative Models and next-noise prediction-based Non-Autoregressive Models, e.g., Diffusion Models. In this work, we propose using discrete-state models to connect them and explore their scalability in the vision domain. First, we conduct an in-depth analysis in a unified design space across two types of models including timestep-independence, noise schedule, temperature, guidance strength, etc in a scalable manner. Second, from the lens of generative models, we re-cast typical discriminative tasks, e.g., image segmentation, as an unmasking process from `[MASK]` tokens on a discrete-state model. This enables us to perform various sampling processes, including flexible conditional sampling by only training once to model the joint distribution. All aforementioned explorations lead to our framework named Discrete Interpolants, which enables us to achieve state-of-the-art or competitive performance compared to previous discrete-state based methods in various benchmarks, including ImageNet256, MS COCO, CC12M, as well as the video datasets FaceForensics and DMLab. In summary, by leveraging `[MASK]` in discrete-state models, we can bridge Masked Generative and Non-autoregressive Diffusion models, as well as generative and discriminative tasks. Our code will be released.

## 1 Introduction

Discrete tokens Esser et al. (2021b); Rombach et al. (2022); Yu et al. (2023) have gained great attention due to their compatibility with LLMs Xie et al. (2024); Zhou et al. (2024) and compactness Yu et al. (2024b); Weber et al. (2024). Based on this, Masked Generative models Chang et al. (2022); Li et al. (2023b) like MaskGiT Chang et al. (2022) have proposed gradually unmasking tokens according to specific heuristically designed rules in the vision domain. Non-Autoregressive Models, e.g., Diffusion Models—especially continuous diffusion models Sohl-Dickstein et al. (2015); Song et al. (2021); Ho et al. (2020); Hu et al. (2024c)—have contributed significantly to the generative community due to their efficacy in score prediction Song & Ermon (2019), conditional synthesis Hu et al. (2023c); Schusterbauer et al. (2024); Gui et al. (2025); Rombach et al. (2022); Hu et al. (2024a), likelihood estimation Song et al. (2021), and image inversion He et al. (2024). As research progresses from continuous-state to discrete-state diffusion models, the training and sampling similarity between Diffusion Models and Masked Generative Models become increasingly noticeable. Yet, a comprehensive analysis of their shared design space and theoretical underpinnings in the vision domain remains conspicuously absent.

To fill this gap, we explore a framework Discrete Interpolants that builds upon the Discrete Flow Matching Gat et al. (2024); Campbell et al. (2024); Shi et al. (2024), which offers flexible noise scheduling and generalization to other methods by considering discrete-state data. While this previous work initially focused on language modeling and explored only the small-scale CIFAR10 vision dataset, we explore the framework to a large-scale realistic dataset. We investigate conventional Explicit Timestep Diffusion models, which explicitly depend on timestep, as well as more flexible Implicit Timestep Diffusion models that completely remove timestep dependence. Additionally, we validate sampling behavior with our framework following the Masked Generative Models approach. This comprehensive investigation deepens our understanding of the connection between Masked Generative Models and Diffusion Models.

On the other hand, there's a trend towards unifying discriminative and generative tasks Grathwohl et al. (2019); Li et al. (2023a); Fuest et al. (2024); Mizrahi et al. (2024); Bachmann et al. (2024). In this work, we demonstrate how to recast the image segmentation task into an unmasking process of our Discrete Interpolants framework. Unlike Mizrahi et al. (2024); Bachmann et al. (2024), which approach the task from the perspective of masked image modeling Hondru et al.

(2024), we instead adopt the perspective of discrete diffusion. In detail, given pairs of image and its segmentation mask, by training them *jointly just once* to model the joint distribution in discrete-state, we can adapt our framework to various discriminative and generative tasks such as image-conditioned semantic segmentation, segmentation mask-conditioned image generation. In summary, our contributions include:

- We abstract and conceptualize various schedulers from discrete flow matching theory, summarizing and generalizing our framework to incorporate different coupling and conditioning methods as special cases. We utilize the progressive generalization from Explicit Timestep Models to Implicit Timestep Models to bring a closer connection between the Diffusion Model and Masked Generative Models.

- We provide an in-depth analysis of the unified design space between Diffusion Models and Masked Generative Models, offering valuable insights for future research. Additionally, we propose that dense-pixel prediction can be reframed as an unmasking process. Furthermore, we present a comprehensive analysis of conditional generation following joint multi-modal training on the Cityscapes dataset.

- Most importantly, by integrating all previous designs, we achieve state-of-the-art performance on the MS COCO and CC12M (12 million images) datasets, while also obtaining competitive results on ImageNet256 and the video datasets FaceForensics and DMLab.

## 2 Related Work

*Due to space constraints, we have included additional related works in the Appendix.*

**Discrete Diffusion Models.** Several works have demonstrated deep connections between diffusion models and auto-regressive models Gat et al. (2024); Ou et al. (2024); Sahoo et al. (2024); Zheng et al. (2024); Liu et al. (2024); Shi et al. (2024). While this has been mainly explored in text generation, in a scaled manner Gong et al. (2024); Nie et al. (2024), we aim to investigate it in the vision domain. Our work differs from most others by exploring a unified design across Masked Generative Models and Diffusion models, providing a more general masking schedule in discrete-state models. MaskGIT Chang et al. (2022), MAGVIT Yu et al. (2024a), Phenaki Villegas et al. (2022), and MUSE Chang et al. (2023) focus on masked generative modeling for generation in a random order (predicting groups of tokens instead of individual tokens), we directly deploy discrete diffusion models on discrete tokens and bring the connection to them through the property of timestep-independence, and provide an in-depth analysis about different aspects of discrete diffusion. VQ-Diffusion Gu et al. (2022) is a discrete-state model specifically designed for vision generation. Unlike it, we further generalize to connect between Masked Generative Model and Diffusion models by utilizing the discrete-state under our Discrete Interpolants framework.

**Connection between Diffusion, and Masked Generative Models.** Most Masked Generative Models Chang et al. (2022); Li et al. (2023b) use heuristically designed, greedy sampling rules based on metrics like purity Tang et al. (2022) or confidence Chang et al. (2022). However, these approaches have been shown to cause over-sampling issues Gat et al. (2024). While our framework leads to a similar training paradigm, it offers a fresh perspective from diffusion models. This introduces new design spaces, including loss weight considerations. Additionally, our sampling approach is more flexible, allowing for both implicit timestep and explicit timestep sampling.

Noticeably, our work is not a direct application of Gat et al. (2024); Shi et al. (2024); Ou et al. (2024); Sahoo et al. (2024); Zheng et al. (2024), though we arrive at a similar conclusion: diffusion models and masked generative models are connected through discrete-state modeling. However, our focus is on scaling this exploration within the image domain, using a large-scale dataset (12M samples from CC12M), which has not been previously validated—e.g., Gat et al. (2024) only considers the CIFAR-10 dataset. Secondly, we establish an additional connection between our method and discriminative approaches, such as segmentation, through the lens of generative models. Lastly, we unify the design space and provide further discussions to strengthen these connections from multiple perspectives.

**Explicit and Implicit Timestep.** Diffusion models Sohl-Dickstein et al. (2015); Ho et al. (2020); Song et al. (2021) are inherently designed with timestep dependence. Several studies have explored the possibility of making timesteps independent Sun et al. (2025). Removing timestep dependence can inherently benefit clean feature extraction Stracke et al. (2025) or enable more flexible sampling Chang et al. (2022). This, in turn, eliminates the need for averaging Fundel et al. (2025) or heuristic search Hu et al. (2023a) when utilizing off-the-shelf features from diffusion models.

| Schedule | Equation $\kappa_t$ | Derivative $\dot{\kappa}_t$ |
|---|---|---|
| Root Weber et al. (2024) | $\sqrt{t}$ | $\frac{1}{2\sqrt{t}}$ |
| Linear Campbell et al. (2024) | $t$ | $1$ |
| Cosine Chang et al. (2022) | $1 - cos(\frac{\pi t}{2})$ | $\frac{\pi}{2}\sin\left(\frac{\pi t}{2}\right)$ |
| Arccos Weber et al. (2024) | $1 - \frac{2 arccos(t)}{\pi}$ | $\frac{2}{\pi\sqrt{1-t^2}}$ |
| Quadratic | $t^2$ | $2t$ |
| Cubic Gat et al. (2024) | $-2t^3 + 3t^2 + b(t^3 - t^2)$ $a(t^3 - 2t^2 + t)$ | $(-6 + 3a + 3b)t^2$ $+(6 - 4a - 2b)t + a$ |
| C Coupling Gat et al. (2024) | $\bar{\kappa}_t \cdot \delta_i(x_t) + \bar{\kappa}_0 \cdot (1 - \delta_i(x_t))$ | $--$ |

Table 1: **Different Masking Schedules $\kappa_t$.** These can be applied in Discrete Interpolants: $p_{t|0,1}(x|x_0, x_1) = (1 - \kappa_t)\delta_{x_0}(x) + \kappa_t \delta_{x_1}(x)$. "C Coupling" refers to Conditional coupling as described in Gat et al. (2024), it can be seen as a token-dependent scheduler, for more details please refer to Appendix.

**Discrete-state diffusion models for discriminative tasks.** Compared to Masked Image Models Hondru et al. (2024), e.g.,4M Mizrahi et al. (2024); Bachmann et al. (2024), which focuses on multimodal tasks but performs unmasking in a single step, we instead propose a progressive unmasking process. This approach unifies the framework for generative and discriminative tasks, offering a novel perspective. To illustrate this, we take semantic segmentation as an example. Liu et al. (2023) apply these models to 3D scenes, Wang et al. (2023) investigates segmentation refinement, and Inoue et al. (2023) employs them for layout generation. We jointly train on image-segmentation mask pairs to model the joint distribution by Discrete Interpolants, enabling both image-conditioned mask generation and mask-conditioned image generation.

## 3 Method

### 3.1 Discrete Interpolants

Our method draws inspiration from discrete-state Diffusion/Flow models Gat et al. (2024); Ou et al. (2024); Sahoo et al. (2024); Zheng et al. (2024) and continuous Stochastic Interpolants Albergo & Vanden-Eijnden (2023); Albergo et al. (2023); Ma et al. (2024a). We aim to extend this interpolant method into discrete-state models with the creation of a flexible and scalable framework for discrete-state modeling. Given the $L$-length real data $x_1 \in \mathbb{R}^L$, the entire possible set is defined as $\mathcal{D} = [K]^L$, where $[K] = 1, 2, ..., K$, and $K$ is the vocabulary size (including an extra [MASK] token). $x_0 \in \mathbb{R}^L$ represents noise, typically tokens filled with mask token [M]. For simplicity, we only consider the single token in our later discussion, as the multi-token can be factorized into the single token, respectively; see Appendix for more details.

Our goal is to learn a transition process based on a vector field $u(x_t, t)$ from $x_0$ (fully masked) to real data $x_1$ by progressively unmasking tokens at each timestep $t$. The key process is a Flow Matching-style probability path interpolated according to the masking schedule $\kappa_t$ [1]:

$$p_{t|0,1}(x|x_0, x_1) = (1 - \kappa_t)\delta_{x_0}(x) + \kappa_t \delta_{x_1}(x), \tag{1}$$

where $\delta_{x_0}(x)$ can be instantiated as $\delta_{[M]}(x)$, and $\delta_x(\cdot)$ is a Dirac delta function indicating whether the element is $x$. $\kappa_t \geq 0$ and $\kappa_0 = 0$ and $\kappa_1 = 1$. There are several choices for the masking schedule $\kappa_t$, e.g., linear, cosine, etc, as shown in Tab. 1.

### 3.2 Training

In the case of continuous-state, given a probability path $p(x_t)$ and a vector field $u(x_t, t)$, the key to ensuring that traversal along the vector field $u(x_t, t)$ can yield the probability transition between $p(x_0)$ and $p(x_1)$ is the Continuity Equation Song et al. (2021). Similarly, in discrete-state modeling, there's a counterpart theory called the Kolmogorov

---

[1]Various terms are used to describe interpolants across different papers, such as "noise schedule" in diffusion models, "interpolants" in Albergo et al. (2023), "protocol" in Shih et al. (2022), and "scheduler" in Gat et al. (2024). For simplicity, we consistently refer to this concept as "scheduler" throughout our paper.

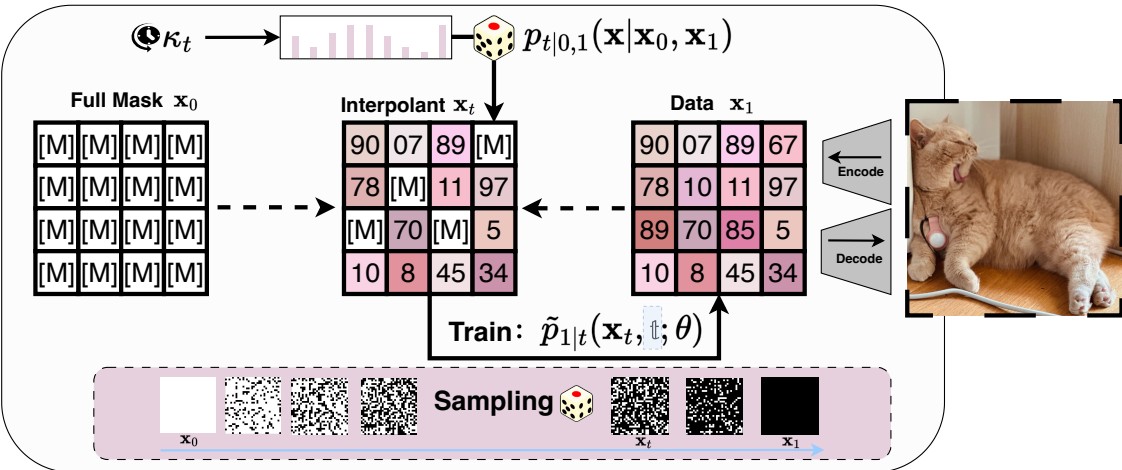

Figure 1: **Discrete Interpolants for training and sampling:** During training, we first obtain discrete interpolants $x_t$ from $x_0$ and $x_1$ following a specific scheduler $\kappa_t$. We then train a model with the cross-entropy loss to predict the original data with $\tilde{p}_{1|t}(\mathbf{x}_t, t; \theta)$, where $t$ indicates that our timestep $t$ is optional, leading to both Explicit Timestep and Implicit Timestep Models. For sampling, we begin with a fully masked $x_0$ and progressively unmask to reach the final fully unmasked $x_1$. Lastly, we decode the indices back to pixel space.

Equation Campbell et al. (2024). It indicates that by following the design of $u_t(x_t) = \frac{\dot{\kappa}_t}{1-\kappa_t}[p_{1|t}(x_1|x_t, t; \theta) - \delta_{x_t}(x)]$, we can traverse along the vector field $u(x_t, t)$ to yield the probability transition between $p(x_0)$ and $p(x_1)$. To learn such a vector field $u_t(x_t) = \frac{\dot{\kappa}_t}{1-\kappa_t}[p_{1|t}(x_1|x_t, t; \theta) - \delta_{x_t}(x)]$, we only need to learn $p_{1|t}^\theta(x_1|x_t, t)$, which incidentally acts as an unmasking function, and can be optimized by a cross-entropy loss:

$$\mathcal{L}(\theta) = \mathbb{E}_{p_{\text{data}}(x_1)p(x_0)\mathcal{U}(t;0,1)p_{t|0,1}(x_t|x_0,x_1)} \log p_{1|t}(x_1|x_t, t; \theta), \tag{2}$$

where $x_t$ is obtained by Discrete Intepolants in Eq. (1).

**Masking and Weighting on Cross-Entropy.** To improve the performance of fidelity, and stabilize the training, we further introduce two significant modifications upon to cross-entropy formulation: a masking operation in the cross-entropy loss and a weighting mechanism:

$$\mathcal{L}(\theta) = \mathbb{E}_{p_{\text{data}}(x_1)p(x_0)\mathcal{U}(t;0,1)p_{t|0,1}(x_t|x_0,x_1)} \left[ \underbrace{w(t)}_{\text{Weighting}} \underbrace{\delta_{[\text{M}]}(x_t)(x_1)^\top}_{\text{Masking}} \log p_{1|t}(x_1|x_t, t; \theta) \right], \tag{3}$$

where $w(t)$ is the weighting function across various timesteps, as indicated by Kingma et al. (2021), the detailed form of $w(t)$ indicates the weighted timestep integral, normally $w(t) = \frac{\dot{\kappa}_t}{1-\kappa_t}$ for the ELBO (Evidence Lower Bound) loss Kingma & Gao (2024), provided the start and end points of the time schedule $\kappa_t$ remain unchanged. However, since ELBO optimization primarily targets improved log-likelihood rather than visual quality, an appropriate $w(t)$ is necessary to enhance visual output. We empirically validate that this holds in the discrete-state setting as well.

**From Explicit Timestep Model to Implicit Timestep Model:** $p_{1|t}(x_1|x_t, t; \theta) \rightarrow p(x_1|x_t; \theta)$. Another interesting property of our method is that $p_{1|t}(x_1|x_t, t; \theta)$ can evolve from an Explicit Timestep Model into an Implicit Timestep Model when considering masked modeling, by removing the dependence of timestep $t$, leading to an unmasking model $p(x_1|x_t; \theta)$. In detail, recent works Ou et al. (2024); Sahoo et al. (2024); Zheng et al. (2024); Shi et al. (2024) have demonstrated that we can remove the dependence on timestep in model design. They show that timestep-dependence can be extracted as a form of weight coefficient outside of the cross-entropy loss. Assume that our scheduler $\kappa_t$ is reversible, even with strict monotonicity, a more intuitive interpretation is that, given a set of masked data by a specific scheduler $\kappa_t$ in $t$, this masked data inherently contains information about which timestep $t$ the data $x_t$ is from (i.e., how

corrupted the data $x_t$ is). Thus, we can safely remove the dependence on the timestep in the model[2]. This implicit timestep property offers several advantages: 1). It establishes a close connection with Masked Generative Model,e.g., MaskGiT Chang et al. (2022), though with a key difference: our discrete model employs independent sampling per token based on pure probability, whereas MaskGit's sampling is heuristically based on the confidence score per token. In experiment, we show that our method can also deploy MaskGit-style sampling but training by our diffusion-based paradigm. 2). The sampling steps can be much simpler and are upper-bounded by the token length $L$. 3). The extra explicit timestep $t$ can sometimes be a restriction in special scenarios, e.g., specific-order sampling or image editing. In these cases, we would prefer to sample auto-regressively or row-by-row. Thus, it's challenging to define the explicit timestep; instead, we could use the implicit timestep from $x_t$ itself. 4). The features extracted from off-the-shelf diffusion models will be cleaner and more suitable for downstream discriminative tasks Stracke et al. (2025).

### 3.3 Sampling

During sampling process, the sample $x_t$ progressively changes between states in $\mathcal{D} = [K]^L$ during sampling, aiming to unmask every token of $x$ with step size $\Delta t$. Our framework offers flexibility in sampling types. We primarily consider three types:

- **Explicit Timestep Model:** This model design is timestep-dependent as conventional diffusion models.

- **Implicit Timestep Model:** This model design is timestep-independent by removing timestep $t$ in the backbone.

- **Masked Generative Model's style sampling** following a greedy heuristic manner: Based on the Implicit Timestep Model, we explore the conventional greedy heuristic style from Masked Generative Models, such as MaskGiT Chang et al. (2022). Our unmasking network functions similarly to Masked Generative Models by recovering masked tokens. We can use our pretrained unmasking network for sampling, following MaskGiT's procedures with $u_t(x_t) = p(x_1|x_t, t; \theta)$ as the token unmasking function.

We detail the sampling process of Explicit and Implicit Timestep Models in Algorithm 1. For the third type, we directly follow MaskGit's pipeline, which we omit from our sampling algorithm, while still using our pretrained timestep-independent unmasking network $u_t(x_t) = p(x_1|x_t; \theta)$.

Notably, unlike the greedy heuristic rule by purity Tang et al. (2022) or confidence Chang et al. (2022), which can guarantee all tokens will be unmasked till the next step, in our model sampled by diffusion models, this is not always true, especially when the sampling step is too small. So we add one extra `argmax` operation on the logits space of the output in the last step of the sampling to mitigate this issue to ensure all tokens will be unmasked. This operation can effectively churn the sampling process to efficiently achieve the data point in a high-fidelity manner.

---

**Algorithm 1 Sampling process of Implicit or Explicit Timestep Model with fixed step size.**

---

**Require:** Network $p(x_1|x_t; \theta)$, masking schedule $\kappa_t$, mask token $m$, time range $t \in [0, 1]$, step size $\Delta t$
    $t \leftarrow 0, x_0 \leftarrow m,$
    **while** $t <= 1 - \Delta t$ **do**
        $u_t(x_t) = \frac{\dot{\kappa}_t}{1-\kappa_t}(p(x_1|x_t, t; \theta) - \delta_{x_t}(x))$
        $p(x_1|x_{t+\Delta t}, t + \Delta t; \theta) \leftarrow \text{Cat}\left[\delta_{x_t}(t + \Delta t) + u_t(x_t)\Delta t\right]$
        $x_{t+\Delta t}^i \sim p^i(x_1|x_{t+\Delta t}, t + \Delta t; \theta)$ for all $x_t^i = m$, $x_{t+\Delta t}^i \leftarrow x_t^i$ for all $x_t^i \neq m$.
        $t \leftarrow t + \Delta t$
    **end while**
    $x_1^i = \text{argmax}\quad u_t^i(x_t)$ for all $x_t^i = m$

---

### 3.4 Segmentation is Unmasking in Discrete Models

A natural extension of Discrete Interpolants is to consider multimodal joint learning the joint distribution, inspired by Assran et al. (2023) as well as recent advances in combined discriminative and generative learning Chen et al. (2024b); Li et al. (2023b). Given real image $x_1 \in \mathbb{R}^{L_x}$ and the second modality $y_1 \in \mathbb{R}^{L_y}$, where $L_x, L_y$ is the token

---

[2]In continuous-state diffusion models, the corrupted data also implicitly includes timestep information, but it's not as straightforward as in discrete-state models to determine the level of corruption in the current data $x_t$.

number of the respective modalities(note here, by default the token dimension of the image should be 2D dimensions, we assume that we have squeezed them into single dimension for better illustration). We simply feed them into the network and obtain their specific logits, therefore, we formulate the unified loss as:

$$\mathcal{L}(\theta) = \mathbb{E}_{p_{\text{data}}(x_1,y_1)\mathcal{U}(t;0,1)p_{t|1}(x_t|x_1)p_{t|1}(y_t|y_1)} \left[ w(t)\delta_{\text{[M]}}(z_t)(z_1)^{\top} \log p_{1|t}(z_1|z_t,t;\theta) \right], \tag{4}$$

where $z_1 = x_1 \oplus y_1$, $z_t = x_t \oplus y_t$, $\oplus$ is the concatenation operation after flattening, and $p_{1|t}(z_1|z_t;\theta)$ is the unmasking process in discrete models by simply replacing real token by [M] according to the masking schedule $\kappa_t$. Noticeably, we share the mask schedule between two different modalities, we find it empirically works well. For the parameterization of $\theta$, the core idea is to standardize each task's inputs and outputs as sequences of discrete vocabulary tokens. To better leverage inductive bias, we retain the patchification approach used in U-ViT Bao et al. (2023). By default, our method operates in latent space to reduce the token count effectively.

Note that while we share similarities with 4M Mizrahi et al. (2024); Bachmann et al. (2024) in using unmasking for segmentation, our focus is establishing connections through the lens of diffusion models, an aspect that has not been explored in-depth in previous works.

### 3.5 Classifier-free Gudiance

We can conduct versatile sampling processes in both single-modality and double-modality scenarios. For the sake of generality, we'll focus on the conditional sampling of $p(x|y;\theta)$ in double-modalities:

$$p(x_1|x_t, y; \omega, \theta) = p(x_1|x_t; \text{[C]}, \theta) + \omega \left[ p(x_1|x_t, y; \theta) - p(x_1|x_t, \text{[C]}; \theta) \right], \tag{5}$$

where $\omega$ represents the guidance strength, and token [C] serves as the null embedding to indicate unconditional signal. By swapping the positions of x and y, we can achieve a similar classifier-free guidance for $p(y|x;\theta)$. This guidance can serve as a plug-in replacement for our previous sampling function described in Algorithm 1.

### 3.6 Extra Discussions

**Conditional Coupling is secretly a token-dependent scheduler.** Conditional Coupling Gat et al. (2024) is a method for coupling $x_0$ and $x_1$ when constructing the discrete interpolants: $x_0 = \mathbb{I}m, \mathbb{I}m, .., (1 - \mathbb{I})x_1, (1 - \mathbb{I})x_1$, it can be seen as a special case of the masking schedule:

$$p_{t|0,1}(x|\bar{x}_0, \bar{x}_1) = (1 - \kappa_t)\delta_{\bar{x}_0}(x) + \kappa_t\delta_{\bar{x}_1}(x), \tag{6}$$

where $(\bar{x}_0, \bar{x}_1) = ([\mathbb{I} \otimes x_1 + (1 - \mathbb{I}) \otimes ([\text{M}], ..., [\text{M}])], x_1)$. It can be understood as a special scheduler built upon the default by making the scheduler dependent on the token's location, $\kappa_t \rightarrow \kappa_t^i$. If $i \notin \mathbb{I}$, it conducts a standard interpolant based on the scheduler; if $i \in \mathbb{I}$, the scheduler is a null scheduler, meaning the token remains unchanged. $\mathbb{I}$ is a mask controlled by a ratio of data length $L$, determining the balance between standard and null schedulers. Noticeably, this coupling is widely applied in text generation, as shown in Gat et al. (2024), to accelerate the model's unmasking process at the final stage of sampling. However, our empirical findings suggest that it is less effective in the vision domain.

**Difference between Discrete Interpolants and Continuous Flow Matching.** Interpolation in the discrete-state setting primarily occurs in the Dirac logits/distribution space $\delta_x(\cdot)$ before sampling from the interpolated logits. Since Dirac distributions are point masses, interpolation in this space involves blending logits rather than directly mixing samples. In contrast, continuous interpolation primarily takes place in the data space.

**Connection between Cold Diffusion Bansal et al. (2024).** Cold Diffusion demonstrates a generalized diffusion model, that can be composed of Degradation Operator $\mathcal{D}$ and Restore Operator $\mathcal{R}$, the degradation can be blurring, pixelated, or snowification, with restriction $\mathcal{D}(x_1, 1) = x_1$, their training is achieved by a minimal optimization problem:

$$\min_{\phi}\mathbb{E}_x||\mathcal{R}(\mathcal{D}(x, t), t; \phi) - x||, \tag{7}$$

our masking operation in Eq. (1) can be naturally a case of the degradation $\mathcal{D}(x_t, t) \sim p_{t|0,1}(x|x_0, x_1)$, and our unmasking network is a case of the Restore Operator $\mathcal{R}(x_t, t)$.

| Model | FID ↓ | Type | Training datasets | #Para. |
|---|---|---|---|---|
| Generative model trained on external large dataset (zero-shot) | | | | |
| LAFITE Zhou et al. (2022) | 26.94 | GAN | CC3M (3M) | 75M + 151M (TE) |
| Parti Yu et al. (2022b) | 7.23 | Autoregressive | LAION (400M) + FIT (400M) + JFT (4B) | 20B + 630M (AE) |
| Re-Imagen Chen et al. (2022) | 6.88 | Continous Diffusion | KNN-ImageText (50M) | 2.5B + 750M (SR) |
| Generative model trained on external large dataset with access to MS-COCO | | | | |
| Re-Imagen‡ Chen et al. (2022) | 5.25 | Diffusion | KNN-ImageText (50M) | 2.5B + 750M (SR) |
| Make-A-Scene Gafni et al. (2022) | 7.55 | Autoregressive | Union datasets (with MS-COCO) (35M) | 4B |
| VQ-Diffusion† Gu et al. (2022) | 13.86 | Discrete diffusion | Conceptual Caption Subset (7M) | 370M |
| Generative model trained on MS-COCO | | | | |
| U-Net | 7.32 | Continuous diffusion | MS-COCO (83K) | 53M + 123M (TE) + 84M (AE) |
| U-ViT Bao et al. (2023) | 5.48 | Continuous diffusion | MS-COCO (83K) | 58M + 123M (TE) + 84M (AE) |
| VQ-Diffusion Gu et al. (2022) | 19.75 | Discrete Diffusion | MS-COCO (83K) | 370M |
| Explicit Timestep Model (*Our*) | **6.03** | Discrete Diffusion & MGM | MS-COCO (83K) | 77M + 123M (TE) + 84M (AE) |
| Implicit Timestep Model (*Our*) | **5.31** | Discrete Diffusion & MGM | MS-COCO (83K) | 77M + 123M (TE) + 84M (AE) |

SR represents a super-resolution module, AE an image autoencoder, and TE a text encoder. Methods marked with † are finetuned on MS-COCO. Those marked with ‡ use MS-COCO as a knowledge base for retrieval.

Table 2: **FID results of different models on MS-COCO** ($256 \times 256$). MGM denotes Masked Generative Models. Baseline results are sourced from U-ViT Bao et al. (2023).

# 4 Experiment

## 4.1 Experimental Details

**Datasets and Metrics.** For image generation, we primarily utilize the ImageNet256 and MS COCO datasets, along with the CC12M dataset, which contains approximately 12 million images Changpinyo et al. (2021), to evaluate scalability. For joint training between image-segmentation mask pairs, we use the Cityscapes dataset. Our video generation experiments mainly utilize the FaceForensics, and more challenging DeepMind Lab (DMLab). We use Fréchet Inception Distance (FID) as the evaluation metric for image generation tasks and CLIP score Hessel et al. (2021) for text-to-image tasks in CC12M dataset. For video generation, we use Fréchet Video Distance (FVD) to assess performance.

**Training Details.** We consistently use the discrete tokenizer SD-VQ-F8 from Stable Diffusion Rombach et al. (2022) across all datasets, due to its extensive pretraining on large datasets like Open-Images Kuznetsova et al. (2020). To avoid singularity issues in the derivative, we sample timesteps $t \in [\epsilon, 1 - \epsilon]$ during training, where $\epsilon = 10^{-3}$. For fair comparison, we employ the same scheduler for both training and sampling, using a fixed step size. For classifier-free guidance, we randomly drop out the conditional signal with a probability of 0.1 during training, following established conventions Ho & Salimans (2021); Hu et al. (2023a). Unlike Gat et al. (2024), which uses adaptive step size for sampling, we consistently employ a fixed step size for fair comparison. Unless otherwise stated, we default to using 1,000 timesteps. For COCO, ImageNet, and Faceforensics, we use linear schedules by default and set $w(t) = 1$, and applying the masking cross-entropy. For more detailed information about the training recipe, optimizer, iteration steps, GPU usage, learning rate, and other parameters, please refer to the Appendix. Code will be released.

## 4.2 Experimental Result

### 4.2.1 Main Result

**Image Generation.** We demonstrate our results on the COCO dataset, as shown in Tab. 2. Our method achieves state-of-the-art performance compared to both continuous-state and discrete-state models. Notably, the Explicit Timestep Model and Implicit Timestep Models perform similarly.

We further showcase our performance on ImageNet256 in Tab. 3. Compared to other conventional Autoregressive and Masked Image Model-based methods, our approach achieves competitive FID scores with around 500M parameters.

**Scalability.** We also evaluate the scalability of our method in 12M scaled dataset CC12M in Tab. 4, our results achieves the best performance compared with previous baselines by using an implicit timestep mode, this demonstrates that our learning paradigm benefits from scaling as autoregressive or diffusion model.

**Video Generation.** To validate our method's scalability, we conduct experiments on the FaceForensics dataset as well as a more dynamic dataset DMLab, as shown in Tab. 5. We adapt the conventional Latte Ma et al. (2024b) models

| Type | Model | #Para. | FID↓ | IS↑ |
|---|---|---|---|---|
| Continuous Diffusion | ADM (Dhariwal & Nichol, 2021) | 554M | 10.94 | 101.0 |
| | CDM (Ho et al., 2022) | – | 4.88 | 158.7 |
| | LDM-4 (Rombach et al., 2022) | 400M | 3.60 | 247.7 |
| | DiT-XL/2 (Peebles & Xie, 2023) | 675M | 2.27 | 278.2 |
| AR & MGM Models | VQGAN (Esser et al., 2021b) | 1.4B | 15.78 | 74.3 |
| | VQGAN-re (Esser et al., 2021b) | 1.4B | 5.20 | 280.3 |
| | ViT-VQGAN (Yu et al., 2022a) | 1.7B | 4.17 | 175.1 |
| | ViT-VQGAN-re (Yu et al., 2022a) | 1.7B | 3.04 | 227.4 |
| | RQTran. (Lee et al., 2022) | 3.8B | 7.55 | 134.0 |
| | RQTran.-re (Lee et al., 2022) | 3.8B | 3.80 | 323.7 |
| | LlamaGen-XL (Sun et al., 2024) | 775M | 3.39 | 227.1 |
| | MaskGIT Chang et al. (2022) | 227M | 6.18 | 182 |
| | Open-MAGVIT2-L Luo et al. (2024) | 804M | 2.51 | 271.7 |
| | MAR Li et al. (2024) | 943M | 2.35 | 227.8 |
| | VAR Tian et al. (2025) | 2B | 1.73 | 350.2 |
| | D-JEPA-H Chen et al. (2024a) | 2B | 2.04 | 239.3 |
| Discrete | VQ-Diffusion Gu et al. (2022) | | 5.32 | – |
| DD & MGM | Explicit Timestep Model | 546M | 3.34 | 286.1 |
| DD & MGM | Implicit Timestep Model | 546M | **3.21** | 283.0 |

Table 3: **Class-conditional generation on ImageNet** $256 \times 256$. MGM denotes Masked Generative Models. AR denotes Auto-Regressive Models. DD denotes Discrete Diffusion Models.

| Method | State | CLIP-Score(10k) ↑ |
|---|---|---|
| LlamaGen Sun et al. (2024)[†] | Discrete | 25.10 |
| Simple Diffusion Hoogeboom et al. (2023) | Continuous | 26.10 |
| MDM Gu et al. (2023) | Continuous | 28.10 |
| Explicit Timestep Model | Discrete | 27.78 |
| Implicit Timestep Model | Discrete | **28.35** |

[†] denotes pretrained by ourself.

Table 4: **Large-scale text-to-image dataset CC12M.**

from continuous-state to discrete-state by inserting a learnable embedding for indices and a linear layer to map the logit dimension from $\mathbb{R}^{W \times H \times T}$ to $\mathbb{R}^{C \times W \times H \times T}$. As demonstrated in the table, our method performs better than its continuous-state counterpart under training settings, indicating that our approach scales from image to video generation. For further information, please refer to the Appendix.

### 4.2.2 Ablation Study

In this section, we analyze the design space of Masked Generative Models and Diffusion Models. Our ablation studies focus on key aspects of both models, including sampling timestep, softmax temperature, classifier-free guidance weight, and the impact of various sampling schedulers. Notably, our model demonstrates the ability to generalize to Masked Generative Model (MGM)-style sampling, further expanding its versatility. For MGM-style models, we also explore a technique called linear Gumbel Noise. This involves linearly adding Gumbel noise to the confidence score in an annealing manner Chang et al. (2022); Besnier & Chen (2023).

**Number of Function Evaluation.** As shown in Fig. S6, the performance of Explicit and Implicit Timestep Models saturates around 100 steps. For MGM-style sampling without Gumbel noise, results are significantly worse across various noise schedules. However, after introducing Gumbel noise, performance saturates at just 10-20 steps. This

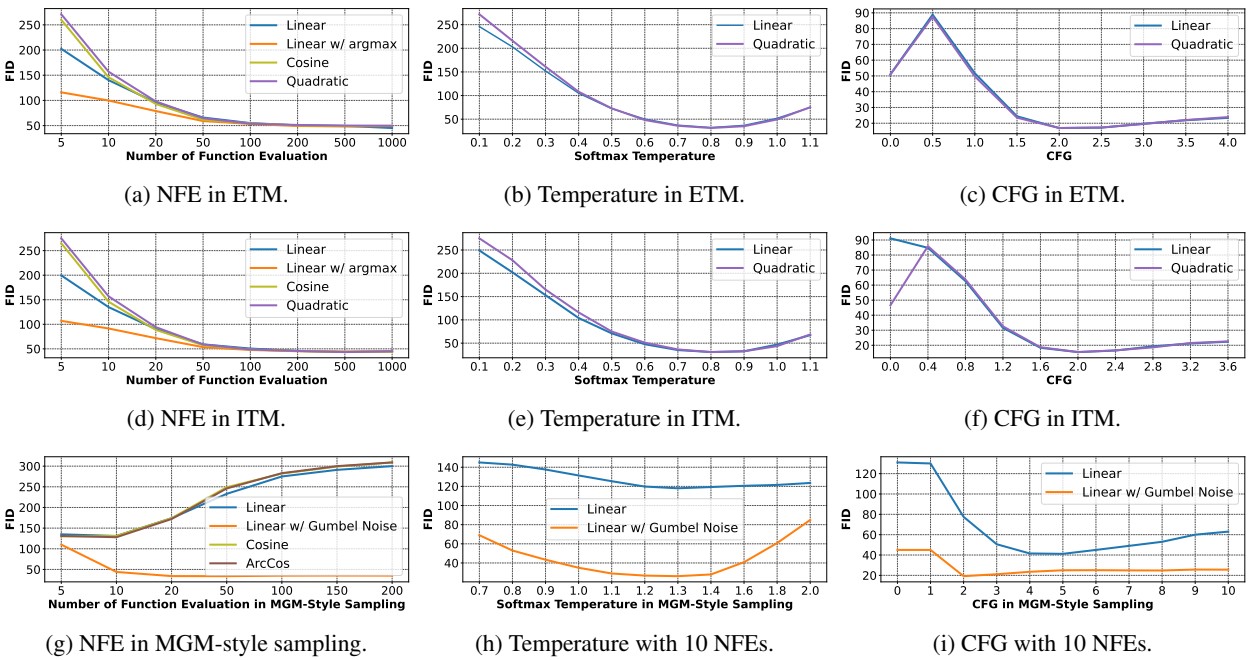

Figure 2: **Ablation about Explicit Timestep Model (ETM), Implicit Timestep Model (ITM), and Masked Generative Model(MGM) style Sampling** on ImageNet 256 dataset with FID-5k. All models are trained with linear schedulers by default.

| Dataset | Method | State | Frame-FID↓ | FVD ↓ |
|---|---|---|---|---|
| FaceForensics | Latte Ma et al. (2024b) | Continuous | 21.20 | 99.53 |
| | Implicit Timestep Model | Discrete | 15.21 | 81.20 |
| | Explicit Timestep Model | Discrete | **13.21** | **77.22** |
| DMLab | Latte Ma et al. (2024b) | Continuous | – | 68.19 |
| | Implicit Timestep Model | Discrete | – | 53.31 |
| | Explicit Timestep Model | Discrete | – | **50.12** |

Table 5: **Video generation result.**

rapid convergence occurs because MGM-style sampling relies on post-sampling confidence, potentially biasing optimal performance towards a lower number of function evaluations (NFE).

**Implicit Timestep vs. Explicit Timestep.** Comparing the first and second rows of Fig. S6, we observe nearly identical optimal FID for low NFE, optimal temperature, and optimal guidance strength of CFG. This similarity persists despite the different training approaches, suggesting that we can safely remove the timestep dependency when necessary.

**Ablate Sampling Scheduler.** From Figs. 2a, 2d and 2g, we observe that the model's optimal FID converges to roughly the same value as we allow longer NFE. However, sampling with the training scheduler (linear) yields better performance. This indicates that misalignment indeed exists across various schedulers.

**Ablating Training Schedule.** We show the ablation of schedule in Appendix. Duet to the significant computation resources requirement of the ImageNet experiment, we ablate it in CIFAR10. We find that a linear schedule generally lead to a better performance compared with other schedulers.

**Softmax temperature.** As seen in Figs. 2b, 2e and 2h, the concept of softmax temperature originates from Masked Generative Models. We observe a sweet spot around 0.8, which interestingly remains consistent regardless of the scheduler choice or whether the timestep-dependence property is implicit or explicit. For MGM-style sampling, the temperature's sweet spot is approximately 1.2, becoming more pronounced after implementing Gumbel noise.

| Method | FID(5k)↓ | mIOU ↑ |
|---|---|---|
| DeepLabv3 Chen et al. (2017) | - | 94 |
| Mask-to-Image Hu et al. (2023b) | 25.4 | - |
| Explicit Timestep Model | 34.4 | 89.1 |
| Implicit Timestep Model | 33.8 | 90.1 |

Table 6: **Ablation results on Cityscapes.**

**Strength of Classifier-free Guidance (CFG).** We also examine the impact of Classifier-free Guidance strength in Figs. 2c, 2f and 2i. Interestingly, the optimal guidance strength remains consistent across different schedulers and timestep properties (implicit or explicit). For MGM-style sampling, we observe that the optimal point shifts to around 3, with the implementation of Gumbel noise yielding superior performance.

**Churning last-step sampling by `argmax`.** To explore how close the sample is to the target distribution, we investigate a technique called argmax. This approach involves using a direct argmax operation on logit space instead of categorical sampling, effectively churning the sampling process with an extremely hard Dirac distribution. As shown in Figs. 2a and 2d, this technique significantly improves sampling performance at low NFE (number of function evaluations).

**Churning entire sampling process by Top-p, Softmax temperature, CFG scale, Gumbel noise.** Top-p (nucleus sampling) and softmax temperature control the overall randomness of the output. In contrast, guidance strength in Classifier-Free Guidance (CFG) steers the sampling process towards specific classes. Gumbel noise adds an extra layer of randomization to the "confidence score" similar to MaskGiT Chang et al. (2022)). As shown in Fig. S6, these techniques can improve performance over the baseline, particularly when using fewer sampling steps (low NFE regimes). Due to space limit, we defer the experiment of top-p in Appendix.

**Image-Segmask Pair Joint Training for Semantic Segmentation.** We conducted our primary experiments on the Cityscapes dataset, as shown in Tab. 6. We also compare with the conventional segmentation method DeepLabV3 Chen et al. (2017), and a typical mask-to-image method Hu et al. (2023c). Our generative models demonstrate versatility by performing both image-conditioned mask generation and mask-conditioned image generation. We evaluated these tasks using FID and mIOU metrics. The results reveal that our framework successfully handles both tasks with a single joint training process. Moreover, the comparable performance of the Explicit and Implicit timestep models suggests that we can leverage the [M] token to reframe discriminative tasks as an unmasking process.

## 5 Conclusion & Future Works

Our Stochastic Interpolants extends discrete flow matching theory to vision tasks, generalizing from Explicit to Implicit Timestep Models. We analyze the intersection of Diffusion and Masked Generative Models, proposing dense-pixel prediction as an unmasking process. By integrating these elements, we achieve state-of-the-art performance on MS-COCO, CC12M, competitive results on ImageNet 256, and demonstrate scalability to video datasets like Forensics, DMLab.

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

## A    Appendix

## B    Potential Impact

Currently, most mask-based methods share a common limitation: once a token is unmasked, it cannot be masked back again, which is also the main motivation of CDCD Dieleman et al. (2022). This means denoising errors can't be reversed or corrected. Liu et al. (2024) propose a solution by decoupling the process into two separate parts, breaking this rule and thus fixing the accumulating error. We anticipate our method can be extended to their paradigm, the smoothing item in our discrete stochastic interpolants can be specifically design for this goal.

One other future work is to consider the modality-dependent masking schedule $\kappa_t$ in multi-modal learning. Our work can be seen as mean-parameterization since it leverages a prediction model for the mean of $x_0$ in a continuous state diffusion model, e.g., DDPM Ho et al. (2020). We anticipate similar to the case of continuous diffusion models, other parameterizations can still yield similar performance.

Despite the underlying conceptual similarities between Masked Generative Models and Diffusion Models, a substantial disparity in sampling efficiency persists between these methods. Discrete Flow Matching Campbell et al. (2024); Gat et al. (2024) typically requires approximately 2,048 sampling steps, whereas MaskGiT Chang et al. (2022) achieves comparable results with merely 18 steps. While the time-independence design in Discrete Flow Matching partially mitigates this gap, the difference in computational efficiency remains significant, highlighting an important area for future research and optimization. The sampling process can be quite flexible by location or probability, for example, we can unmask the token who is maximum probability named confidence score Chang et al. (2022), or purity sampling in Tang et al. (2022); Campbell et al. (2024).

## C    Extra Details

### C.1    Training Detail

We follow the implementation of MaskGiT Chang et al. (2022) and do not use label smoothing or soft target cross-entropy. Empirically, we find that initializing the final fully connected layer of the logits is crucial for stabilizing training, especially in multi-node setups. Gradient clipping with a norm of 2 is applied throughout the training process. Wherever applicable, we utilize fused kernels, such as fusedAdam, to optimize performance.

In our joint training experiments, we explore various configurations of timesteps and shared vocabularies. Our findings indicate that using different timesteps with separate vocabularies consistently yields the best results.

To optimize batch size, we pre-extract the indices in advance. This approach effectively doubles the batch size in many cases. For example, on the MS COCO dataset, the maximum batch size for U-ViT2-S2-Deep on an A100-40G GPU is 28 without pre-extraction, but increases to 48 with pre-extracted indices. Additionally, in joint training, pre-tokenizing accelerates training speed significantly, improving from 1.3 iterations per second to 4.2 iterations per second with a batch size of 16.

For more training details, please refer to Tab. S7.

### C.2    Evaluation Detail

For the ImageNet256 experiment, we follow the same evaluation protocol and implementation as ADM. For COCO, we validate the FID using the validation set. For FVD, we adhere to the evaluation guidelines outlined by StyleGAN-V Skorokhodov et al. (2022), computing FVD scores on 2,048 video clips, each consisting of 16 frames.

Notably, we do not apply corrector sampling, as used in Discrete Flow Matching, since such techniques are typically employed with larger models, such as those with 1.7B parameters.

We do not consider C Coupling in our method, as it was proposed in Discrete Flow Matching for text generation. Typically, C Coupling requires that partial data consists of non-masked tokens rather than fully masked data. However, we empirically find that this leads to a misalignment between training and sampling. Since our starting point is always fully masked, which is not always the case in code generation, we choose not to adopt C Coupling.

| | ImageNet256 | Cityscapes | MS-COCO | FaceForensics |
|---|---|---|---|---|
| Raw Input shape | $3 \times 256 \times 256$ | $3 \times 256 \times 256$ | $3 \times 256 \times 256$ | $16 \times 3 \times 256 \times 256$ |
| Discrete indices shape | $32 \times 32$ | $32 \times 32$ | $32 \times 32$ | $16 \times 32 \times 32$ |
| Tokenizer | SD-VQ-F8 | SD-VQ-F8 | SD-VQ-F8 | SD-VQ-F8 |
| Vocab size | 16,385 | 16,385 | 16,385 | 16,385 |
| Optimizer | AdamW | AdamW | AdamW | AdamW |
| Learning rate | 2e-4 | 1e-4 | 2e-4 | 1e-4 |
| Weight decay | 0.00 | 0.00 | 0.0 | 0.00 |
| #Param(M) | 546 | 38 | 77 | 764 |
| Training iterations | 500K | 400K | 1M | 500K |
| Warm-up steps | 5K | 5K | 5K | 0K |
| Batch size | 1,024 | 128 | 1,024 | 128 |
| GPU | 64 | 8 | 64 | 128 |
| Local batch size | 16 | 16 | 16 | 1 |
| Training Time (hours) | 48 | 48 | 24 | 48 |
| Sampling steps-ITM/ETM | 20/1k | 20/1k | 20/1k | 20/1k |
| top-p | 0.9 | 0.9 | 0.9 | 0.9 |
| Softmax temperature | 1.3 | 1 | 0.7 | 1 |
| Gumbel noise temperature | 4.5 | 4.5 | 4.5 | 4.5 |
| CFG | ✓ | ✓ | ✓ | ✗ |
| $p_{\mathrm{uncond}}$ | 0.1 | 0.1 | 0.1 | – |
| Guidance strength | 2 | 2 | 4.5 | – |

Table S7: **Training Detail.**

## C.3 Dataset Detail

**ImageNet256, MS COCO and Cityscapes.** We use ImageNet256 for ablation study and main comparison following previous works Bao et al. (2023). We use MS COCO 2014 Lin et al. (2014) to explore the alignment between images and captions. For the evaluation of FID on COCO $256 \times 256$, we sample prompts from the validation set, following the approach of U-ViT Bao et al. (2023). We use the Cityscapes dataset to explore the alignment between images and segmentation. Following Hoogeboom et al. (2021), we reduce the computational cost by rescaling the segmentation maps from Cityscapes to $32 \times 64$ images using nearest neighbor interpolation. We utilize the global categories as prediction targets, resulting in an 8-class problem. For the real images used in FID computation, we randomly crop $256 \times 256$ images from the $256 \times 512$ landscape Cityscapes images.

**CC12M.** CC12M is a dataset consisting of approximately 12 million image-text pairs, designed for vision-and-language pre-training. As mentioned earlier, we select CC12M as our primary training set due to its moderate size, which is well-suited for developing high-quality text-to-image models with strong zero-shot capabilities. Additionally, the dataset is freely available and poses fewer privacy concerns.

In this paper, we utilize the entire set of text-image pairs for text-to-image generation. Specifically, we randomly sample 1/1000 of the pairs as a validation set, where we track CLIP and FID scores during training, while the remaining data is used for model training. By default, each image is center-cropped and resized to the required resolution based on the task. No additional filtering or cleaning is applied.

**FaceForensics.** FaceForensics contains 150×150 images of deepfake faces, totaling over 20,000 images from 1,000 videos. For video generation, we use an image-based tokenizer: SD-VQGAN Rombach et al. (2022). We extract 16-frame video clips from these datasets using a specific sampling interval. Each frame is resized to 256×256 resolution for training, resulting in a size of 16×3×256×256. For evaluation of video generation, we follow the code of previous works Ma et al. (2024b); Skorokhodov et al. (2022) for FVD calculation.

# D  Extra Discussions

## D.1  Scheduler

**Smoothing factor.**    Sometimes, the transition between the probability distribution $p(x_0)$ and $p(x_1)$ by such a binary interpolation of $\kappa_t, 1 - \kappa_t$ is too rigid, to alleviate this rigid issue, a smoothing factor $\gamma_t$ is introduced as:

$$p_{t|0,1}(x|x_0, x_1) = (1 - \kappa_t)\delta_{x_0}(x) + p^{smooth}(x) + \kappa_t \delta_{x_1}(x), \tag{8}$$

where $\delta_{x_0}(x)$ can be instantialized as $\delta_{[\text{M}]}$ and $\kappa_t \geq 0$. Intuitively, the smoothing term can bring two benefits: 1) data augmentation on the clean data. 2), it provides extra training for the corrector sampler. A common instantiation can be $\kappa_0 = 0$, $\kappa_1 = 1$, and $p^{smooth}(x)$ is a uniform distribution: $p^{smooth}(x) \sim U(0, 1)$. To sample from $p^{smooth}_{t|0,1}(x|x_0, x_1)$, the sample $x_t$ can be obtained by from $\sqrt{s \cdot \kappa_t \cdot (1 - \kappa_t)}\mathbb{I}$, where $\mathbb{I}$ is an $\mathbb{R}^{L \times K}$ tensor full of one, s is used to control the stochasticity in the sampling process. The insight of the adding such a smoothing factor, is that, it encourages the interpolants between other uninvolved "word" in the $k$-sized vocabulary, it's similar to the Brownian Bridge instantiation Tong et al. (2023); Albergo et al. (2023). If $s = 0$, it will degenerate to simple discrete interpolants of Eq. (1), if $s \to +\infty$, then the interpolants is over-smoothed, so that we are nearly sampling from a uniform distribution, which is not desired. Another interesting interpretation is to see it as data smoothing, with as a counterpart of label smoothing or data mixup Zhang et al. (2017); Chen et al. (2020).

Uniform noise Campbell et al. (2024), is a special case of our discrete stochastic interpolants. It injects non-[M] tokens to noise the real token, similar to the goal of our smoothing factor. This design offers two benefits: it acts as a form of data augmentation and improves training for corrector sampling, allowing for the possibility of masking back data tokens—a topic beyond the scope of our paper.

This Conditional Coupling is designed to encourage the network to better unmask the data when encountering a specific ratio of masked data. As previously mentioned, discrete data inherently contains timestep information, potentially causing misalignment between the mask ratio and the timestep. Fortunately, the network can be designed in an implicit timestep manner. By utilizing this time-independence design, the benefits of the conditional coupling scheduler can be better leveraged.

**Why do we need various schedulers?**    Assuming we have L-dimensional data, the number of possible masking cases is $C_L^1 + C_L^2 + ... + C_L^L$, which approaches infinity as L grows large. For example, in text, L represents the number of tokens, while in computer vision, it could be 1,024 tokens under the LDM tokenizer for a $256 \times 256$ image. During training, it's impossible to enumerate all possible cases sufficiently. Therefore, we need to design various schedulers to facilitate the learning process. This suggests that changing the schedulers in the sampling process from those used in training will typically decrease performance, as we show empirically in our experiments. However, since the overall learning paradigm remains identical—with the masking schedule being the only difference—we demonstrate that fine-tuning with target samplers using minimal tokens can achieve performance similar to that of source schedulers.

One extra benefit of the discrete stochastic interpolates, it enables better sampling performance, e.g., In MaskGiT, To prevent MaskGIT from making overly greedy selections, random noise is added to the confidence scores, with its magnitude annealed to zero following a linear schedule.

In this way, we can uniformly cast these discriminative tasks as an unmasking process in discrete-state modeling. Classification and semantic segmentation have possibility sets of $[1]^K$ and $[L]^K$ respectively. This recasting enables us to better utilize the discrete nature of segmentation and pixel-level classification tasks.

## D.2  Connections with Other Methods

**Connection between BERT Kenton & Toutanova (2019) and MAE He et al. (2022).**    BERT conducts masked training on data, feeding the masked tokens into the encoder network, while MAE removes these tokens from the encoder input. Our method shares similarities with their masking operations but differs in several ways: 1). In training, our masking operation is based on the masking schedule $\kappa_t$, whereas theirs uses a fixed probability. 2). In sampling, our unmasking operation is conducted progressively, while BERT and MAE unmask all tokens at once. 3). Our masking operation extends to other possible modalities, enabling more flexible sampling. To better illustrate the connection, we represent their masking schedule using our notation in a timestep-independent manner:

$$\kappa_t = d, 1 - \kappa_t = 1 - d, \tag{9}$$

where $\kappa_t$ is always timestep-indepdent, and $d$ is the fixed probability of masking *real* data.

### D.3 From Single-Token to Multi-Token

In main paper, we mainly consider single token and introduce the interpolants, and training process of them. In real scenario, data $x$ can be a $L$-length token sequence $x^i, i \in [L]$. We assume the interpolants operation can be factorized as token-wise. The network predicts the probabilities of all tokens at a time $t$. The loss in Eq. (3) under the multi-token case can be written as:

$$\mathcal{L}(\theta) = \mathbb{E}_{p_{\text{data}}(x_1)p(x_0)\mathcal{U}(t;0,1)p_{t|0,1}(x_t|x_0,x_1)} \left[ w(t) \sum_{l:x_t^l = [\text{M}]} (x_1^l)^\top \log p_{1|t}^l(x_1|x_t,t;\theta) \right]. \tag{10}$$

Given that sampling is independent for each token, and interactions between tokens occur only within the network parameterized by $\theta$. This shares the same expression as previous works Shi et al. (2024); Gong et al. (2024) with masking and weighting. We'll express the sampling using the formulation of the single-token case, omitting multi-token considerations (the superscript $l$), to enhance clarity and understanding.

## E   Extra Related Works

### E.1 Discrete and Continuous Representation

The debate between discrete and continuous representations in generative models, as explored in works like GIVT Tschannen et al. (2024) and MAR Li et al. (2024), has highlighted their respective strengths. However, these paradigms are not mutually exclusive. For instance, increasing the token vocabulary (e.g., using a large codebook of 162K can yield similar benefits to continuous-valued tokens, as shown in GIVT Tschannen et al. (2024). This suggests that discrete representations can bridge the gap to continuous approaches under certain conditions.

Discrete-valued tokens offer several compelling advantages: (1) compatibility with large language models (LLMs)Xie et al. (2024), (2) efficient compression for edge devices, (3) improved visual understandingGe et al. (2024), and (4) enhanced robustness in vision tasks Mao et al. (2021). Recent advances, such as LlaMaGen Sun et al. (2024), have demonstrated that discrete tokenizers can be competitive with continuous latent space representations. Prominent examples include SD VAE Rombach et al. (2022), SDXL VAE Podell et al. (2024), and OpenAI's Consistency Decoder, all widely adopted in diffusion models. These developments indicate that discrete representations in image tokenizers are no longer a bottleneck for image reconstruction. However, latent diffusion models Rombach et al. (2022) have established that continuous representations still hold an edge in certain scenarios.

Discrete-state generative models can be categorized into two main types: (1) next-token prediction Sun et al. (2024) and (2) next-set-of-token prediction Chang et al. (2022). These models aim to: (1) determine the sequence in which tokens are predicted (unmasked) and (2) identify the token to predict at each position using a score such as purity Tang et al. (2022), confidence Chang et al. (2022), or pure sampling Gat et al. (2024).

In exploring the potential of discrete visual tokens Esser et al. (2021b); Zha et al. (2024), works such as VQ-GAN Esser et al. (2021b), VQ-VAE Van Den Oord et al. (2017), GSQ Wang et al. (2024), and MAGVIT-v2 Yu et al. (2024a) have laid a strong foundation. Building on these efforts, we further investigate discrete diffusion models by leveraging pretrained discrete tokenizers. Recent work, such as Hu et al. (2024b), also explores intermediate learned embeddings while relying on continuous-state-based theories.

### E.2 Autoregressive and Non-autoregressive Models

The concept of masking aligns with several other works Austin et al. (2021); Gat et al. (2024); Ou et al. (2024); Campbell et al. (2024). While these primarily focus on linear schedulers, our approach generalizes it to more flexible scheduling options.

Unified-IO Lu et al. (2022) shares our goal of modeling joint distributions across multiple modalities. However, their approach applies a raw transformer to attend to different modalities without considering masked diffusion models.

| Method | Timestep Dependence | FID5k ↓ |
|--------|:---:|---------|
| Cubic | ✓ | 33.46 |
| Cosine | ✓ | 41.75 |
| C Coupling | ✓ | 24.04 |
| Linear | ✗ | 20.67 |
| Linear | ✓ | 20.05 |

Table S8: **Ablation of Training Schedule of CIFAR-10 dataset.**

Unified-IO represents all image-like modalities using a pre-trained RGB VQ-GAN Esser et al. (2021b), whereas we adopt modality-specific tokenizers tailored to each modality. Similarly, ImageBart Esser et al. (2021a) amortizes the effort of handling different timesteps into separate models. In contrast, we consolidate this effort into a single model, avoiding the complexity of designing separate weights for each timestep.

On the other side, there is a lot of exploration between the combination of Autoregressive and Non-Autoregressive models, e.g., Show-o Xie et al. (2024), Transfusion Zhou et al. (2024). Show-o rephrases the problem from the perspective of MaskGit Chang et al. (2022), and they don't have any concept about timestep in multi-stage training, but we aim to solve the problem from the perspective of discrete interpolants on a single stage and explore it around various noise schedules. Additionally, we extend the scope to encompass image generation, segmentation, and video generation. Transfusion Zhou et al. (2024), considers mixed state with discrete representation from text, and continuous representation from images, with dual loss from language and diffusion losses. Although discrete-state model closely connects the Masked Generative Models and Diffusion Models, and Kilian et al. (2024) tries to analyze the computation across different methods, we aim to provide a systematic analysis of the unified design space between Masked Generative Models and Diffusion Models.

Chameleon Team (2024) introduces a family of token-based mixed-modal models capable of both comprehending and generating images. This approach represents all modalities as discrete tokens and utilizes a unified transformer-based architecture. The model is trained from scratch in an end-to-end manner for autoregressive modeling of visual generation. Our approach differs from this method.

For more details, we encourage readers to refer to the survey Xiong et al. (2024).

### E.3 Implicit and Explicit Timestep in Diffusion Models

Diffusion Models and Feature Representation interact across various domains Fuest et al. (2024); Fundel et al. (2025); Yu et al. (2024c). In most scenarios, diffusion features are extracted in a timestep-dependent manner—either through averaging Fundel et al. (2025) or heuristic search Hu et al. (2023a). We extend this concept to develop Implicit Timestep diffusion models that incorporate timestep dependence within the models for discrete states. This approach is intuitive since masked image tokens inherently contain timestep information (i.e., masking ratio). While timestep independence should be possible in continuous states, limited research exists in this area. We believe this is due to network architecture limitations in detecting subtle changes in continuous time steps-based corruption of the original images. Nevertheless, several studies Stracke et al. (2025) demonstrate that networks can incorporate timestep dependence through fine-tuning, suggesting promise for implicit timestep models.

## F Extra Results

### F.1 Basic Visualization

**Chain visualization of ImageNet and Cityscapes.** To further validate the efficacy, we demonstrate the results of ImageNet256 generation in Fig. S3. We can already generate visually pleasing conditioned images, especially at low NFE, we also try to directly apply the `argmax` on the logit space, we can surprisingly obtain the image close to the final sampled images, which indicates that churning sampling can make the sampling more efficient.

### F.2 More Visualization

Sample visualizations from ImageNet and COCO datasets are shown in Fig. S5 and Fig. S4 respectively.

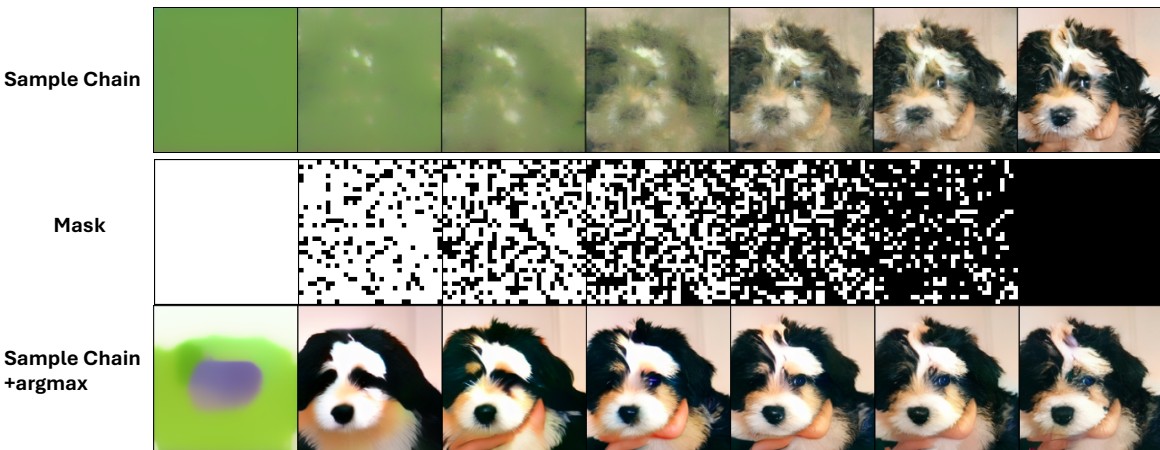

Figure S3: **Chain visualization** for ImageNet 256 with 100 timesteps with `argmax` applied.

In Fig. S6, we provide visualizations comparing different CFG scales, softmax temperatures, Gumbel noise styles, and Gumbel noise temperatures.

We visualize various schedulers in Fig. S7, with their corruption processes shown in Fig. S8.

For the weighting $w(t)$, we showcase its relationship with both time $t$ and signal-to-noise ratio $\text{SNR}(t)$ in Fig. S9.

Our ablation study of top-p sampling on the ImageNet dataset in Fig. S10 reveals that top-p=0.9 yields optimal results.

We demonstrate different factors of the Implicit Timestep Model in Fig. S11.

For Cityscapes dataset, we demonstrate segmentation mask-conditioned image generation in Fig. S12, achieving visually pleasing results with relatively few function evaluations (NFE).

For joint training on the Cityscapes dataset, we visualize discrete token prediction accuracy and loss in Fig. S15. The loss and accuracy patterns are similar between image and segmentation mask generation. However, the mask's cross-entropy loss shows greater stability than the image's loss. The higher accuracy for mask generation indicates it is an easier task than image generation.

**`Argmax` alleviates scheduler misalignment and improves low-NFE performance.** As shown in Fig. S18, our model is trained with a linear scheduler. When we shift the sampling scheduler from linear to cosine, some tokens remain unmasked. Directly applying the `argmax` operation yields a reasonable result, compensating for this issue. This technique also improves performance when sampling at a low NFE, fully unmasking the remaining `[M]` tokens. Interestingly, when we apply `argmax` in the generative chains, it still produces samples which are almost identical to the ground truth.

## G  Licences

### G.1  Datasets

- ImageNet Deng et al. (2009): CC BY 2.0 license

- MS-COCO Lin et al. (2014): Creative Commons Attribution 4.0 License

- CC12M Changpinyo et al. (2021): unknown

- FaceForensics: MIT license

### G.2  Pretrained models

- Image autoencoder from Stable Diffusion Rombach et al. (2022): CreativeML Open RAIL-M License

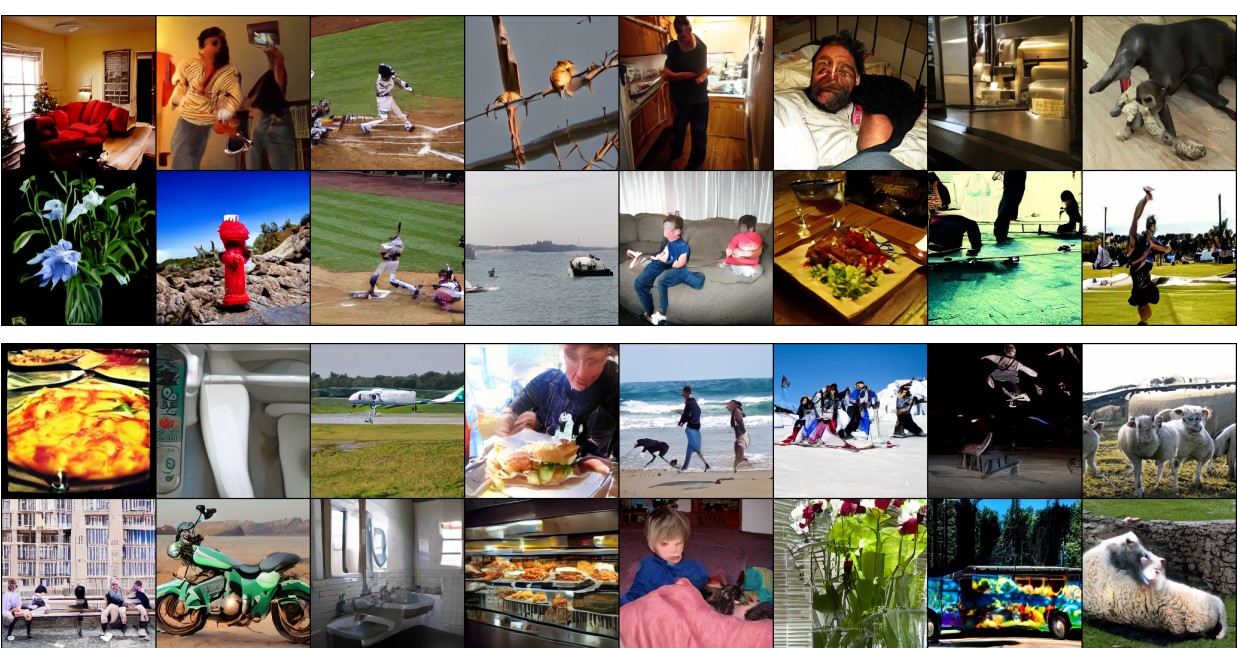

Figure S4: **Non cherry-picked visualization of MS COCO dataset.** CFG=4.5, FID-50k=5.31.

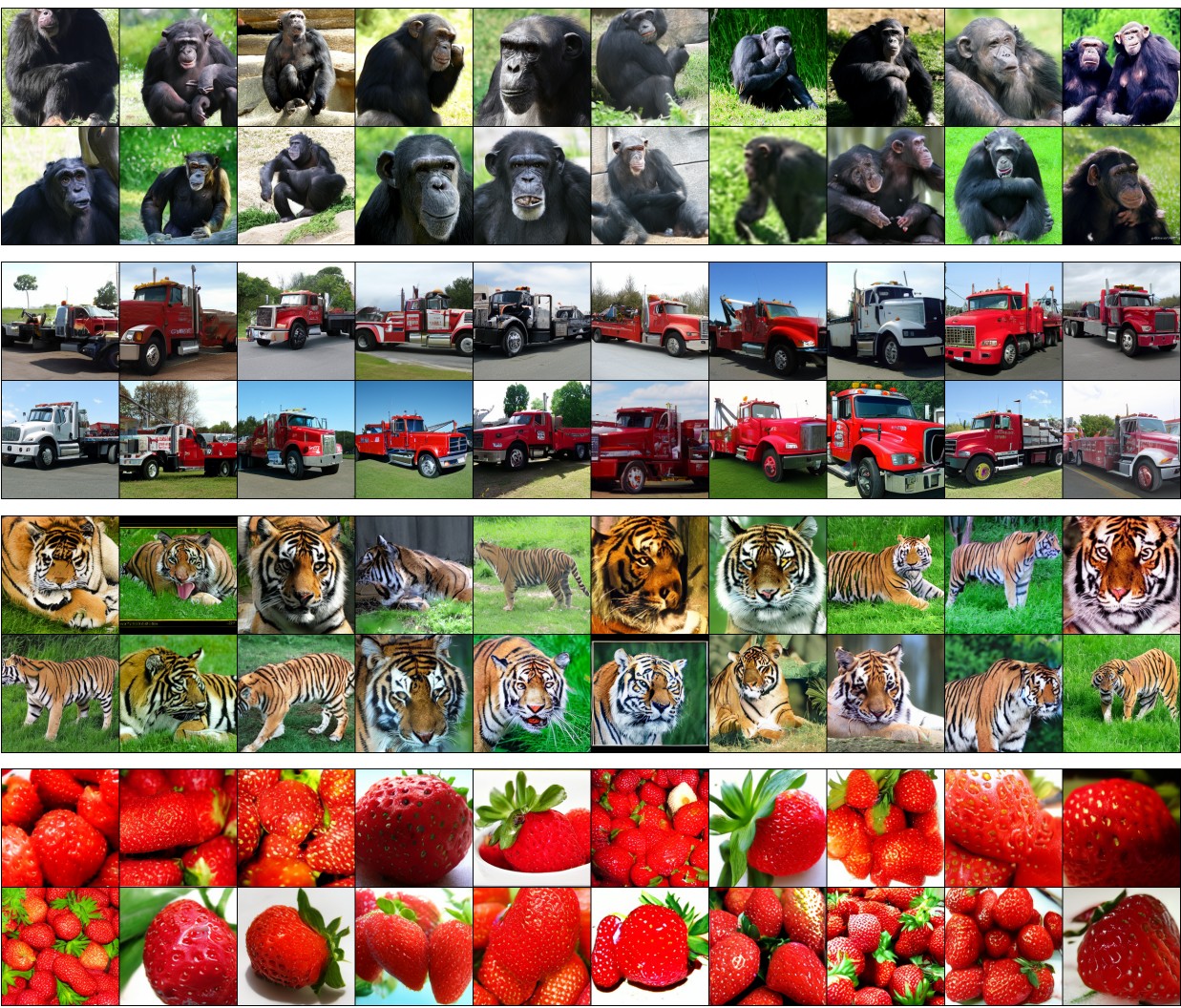

Figure S5: **Non cherry-picked visualization of ImageNet256**. We sample for 20 steps with CFG=3.0, and temperature=1.3.

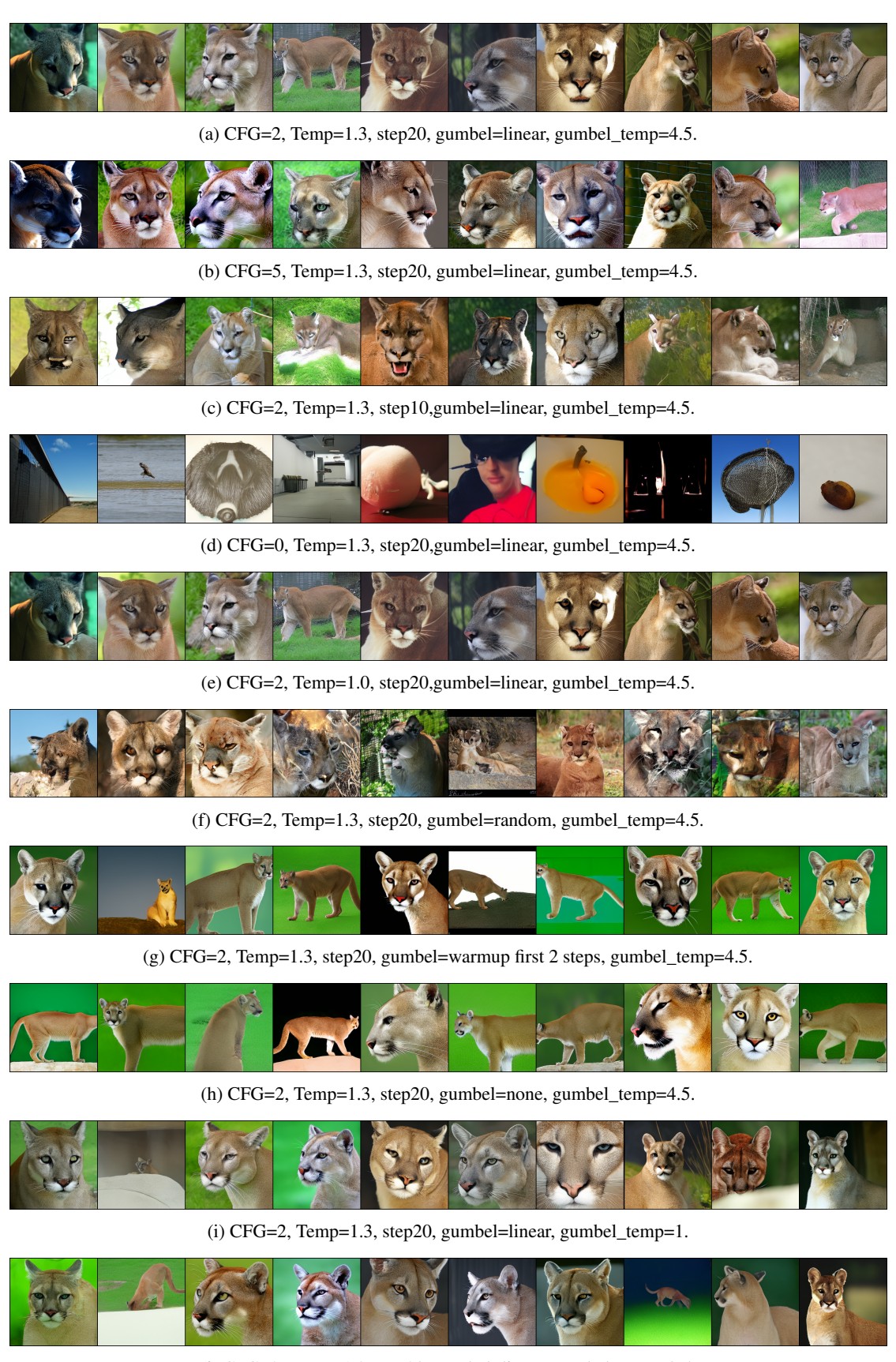

(a) CFG=2, Temp=1.3, step20, gumbel=linear, gumbel_temp=4.5.

(b) CFG=5, Temp=1.3, step20, gumbel=linear, gumbel_temp=4.5.

(c) CFG=2, Temp=1.3, step10, gumbel=linear, gumbel_temp=4.5.

(d) CFG=0, Temp=1.3, step20, gumbel=linear, gumbel_temp=4.5.

(e) CFG=2, Temp=1.0, step20, gumbel=linear, gumbel_temp=4.5.

(f) CFG=2, Temp=1.3, step20, gumbel=random, gumbel_temp=4.5.

(g) CFG=2, Temp=1.3, step20, gumbel=warmup first 2 steps, gumbel_temp=4.5.

(h) CFG=2, Temp=1.3, step20, gumbel=none, gumbel_temp=4.5.

(i) CFG=2, Temp=1.3, step20, gumbel=linear, gumbel_temp=1.

(j) CFG=2, Temp=1.3, step20, gumbel=linear, gumbel_temp=0.5.

Figure S6: **Ablation about different sampling with the same class and same seed.** We mainly compare with different CFG scales, softmax temperature, gumbel noise style, and the temperature of the gumbel noise.

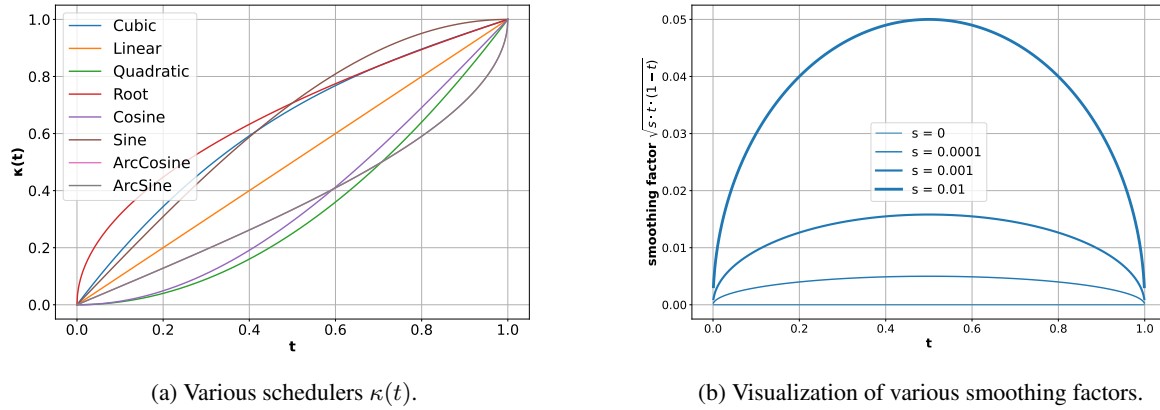

(a) Various schedulers $\kappa(t)$.

(b) Visualization of various smoothing factors.

Figure S7: **Scheduler Visualization.**

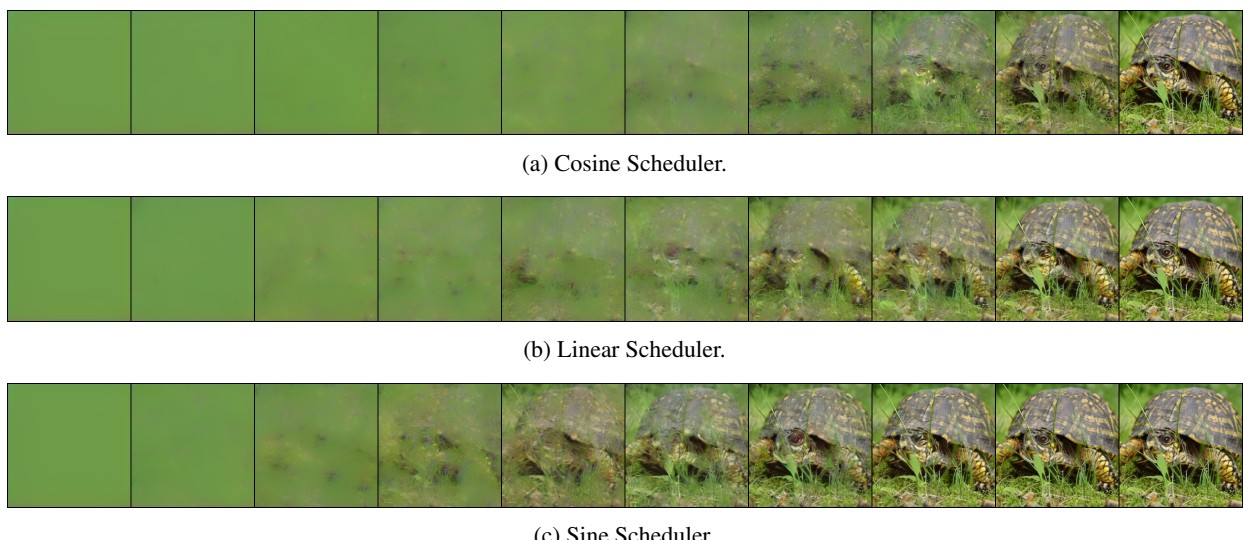

(a) Cosine Scheduler.

(b) Linear Scheduler.

(c) Sine Scheduler.

Figure S8: **Ablation about different schedulers, we mainly consider cosine, linear, and sine schedulers.**

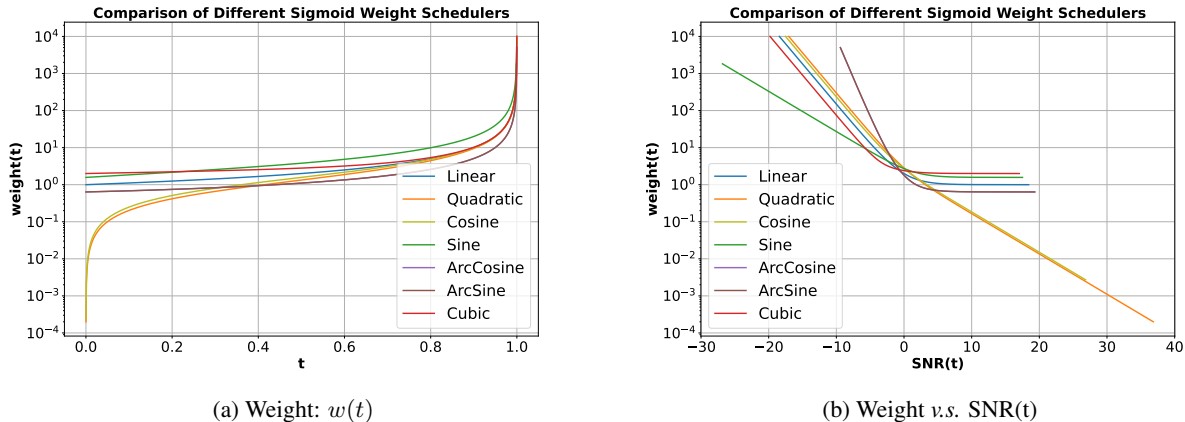

(a) Weight: $w(t)$

(b) Weight *v.s.* SNR(t)

Figure S9: **Weight $w(t)$ visualization.**

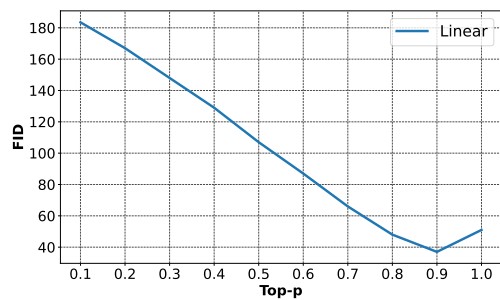

Figure S10: **Top-p v.s FID for ETM** for ImageNet.

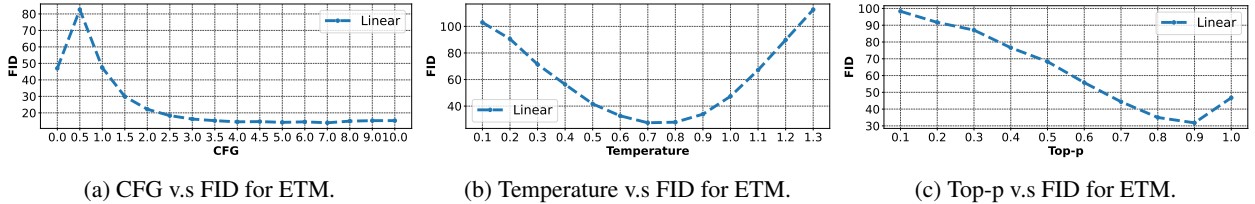

(a) CFG v.s FID for ETM.  (b) Temperature v.s FID for ETM.  (c) Top-p v.s FID for ETM.

Figure S11: **The ablation of CFG,temperature,top-p of ITM(Implicit Timestep Model) in COCO dataset.**

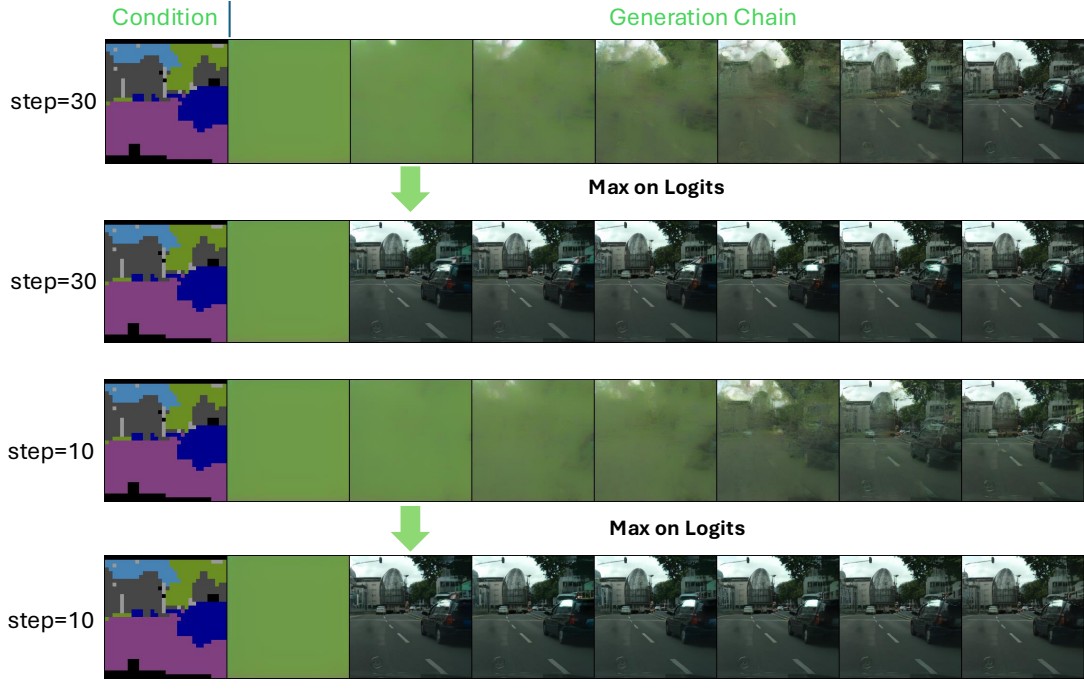

Figure S12: **Mask-conditioned image generation.**

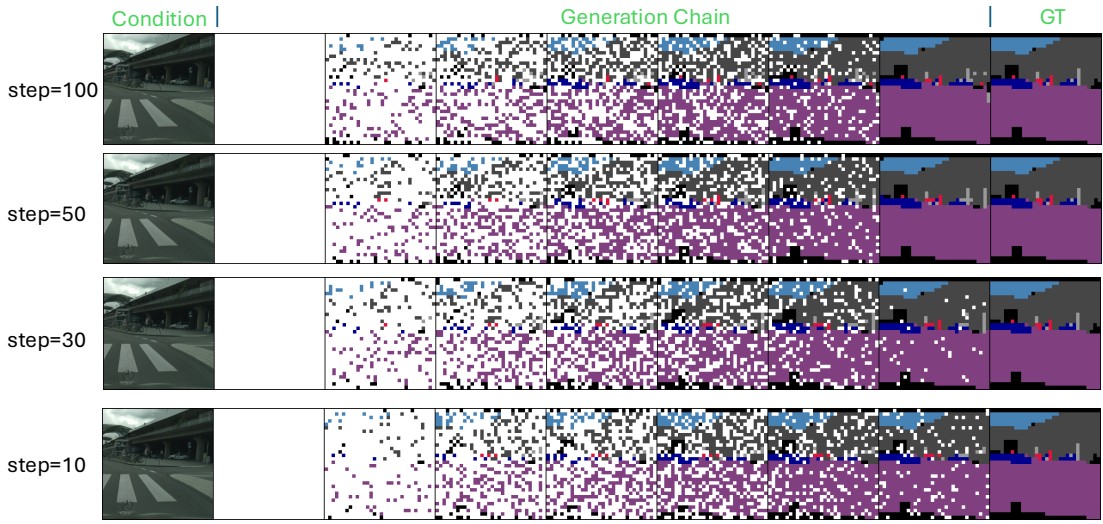

Figure S13: **Progressive chain visualization for different steps** in image-conditioned segmask generation on Cityscapes datasets. We use CFG scale=3.

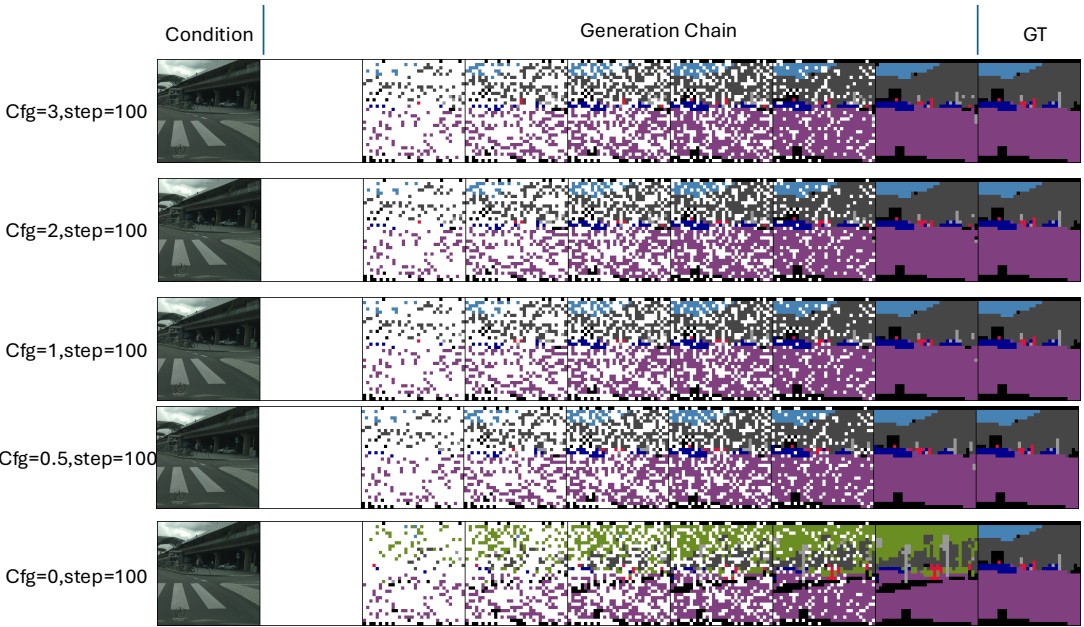

Figure S14: **The guidance scale of classifier-free guidance** for image-conditioned segmask generation on Cityscapes.

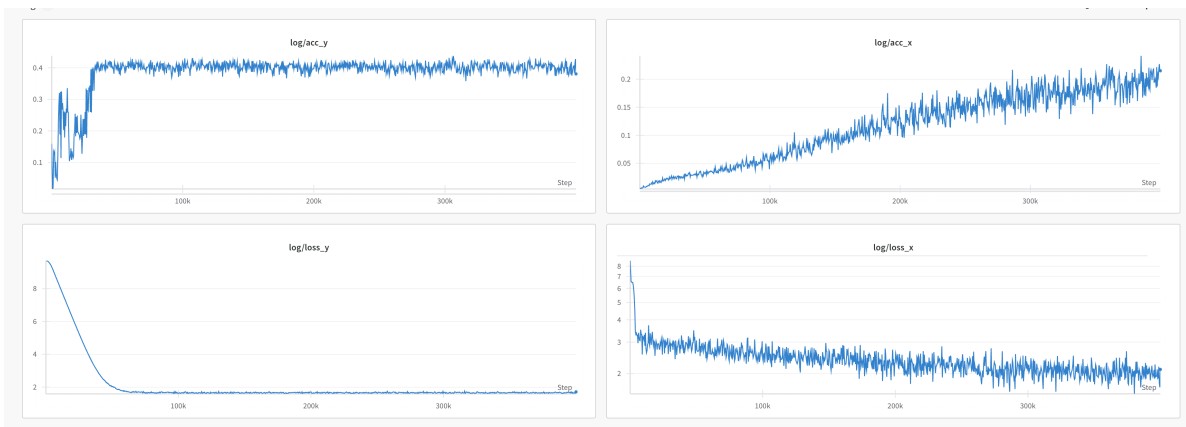

Figure S15: **The loss and accuracy trend of the joint Cityscapes training.** x denotes the image, and y denotes the segmentation mask.

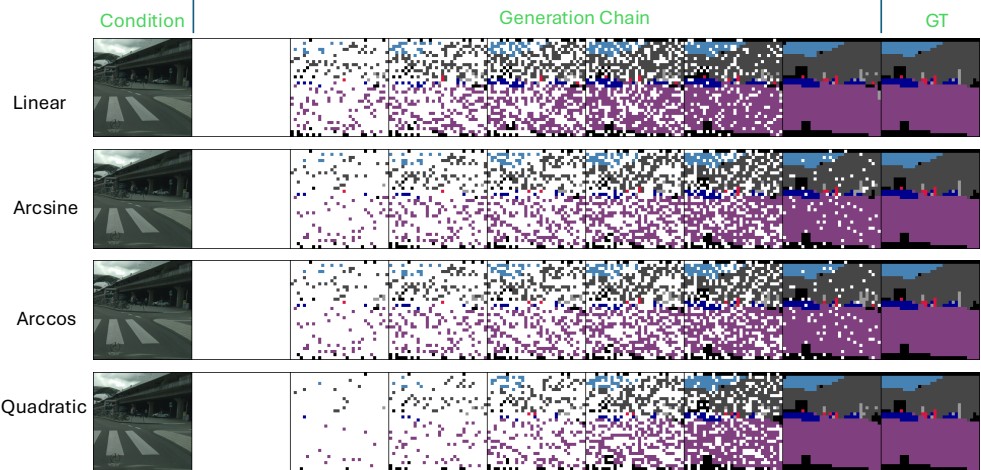

Figure S16: **Misalignment between schedulers. The progressive chain visualization of changing** when the sampling scheduler $\kappa(t)$ when trained with linear schedulers. We sampled with 50 steps and CFG scale=3.

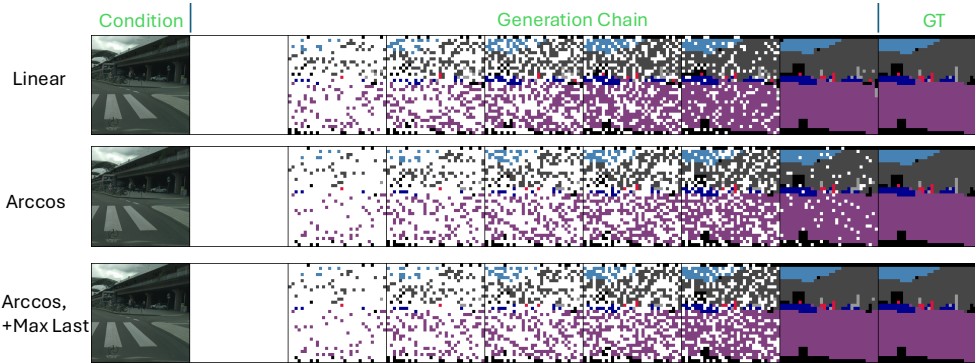

Figure S17: **`argmax` operation after logits can greatly alleviate the issue of those misalignments between training and sampling schedulers.**

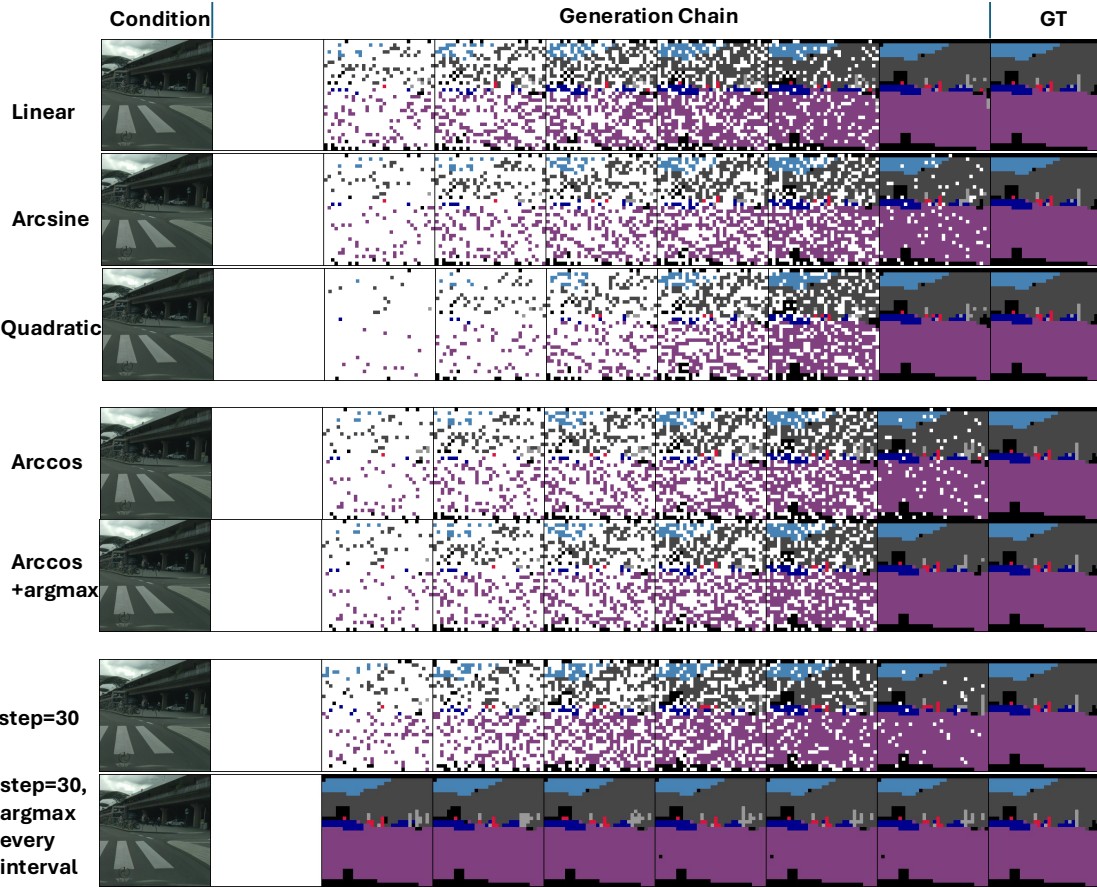

Figure S18: **Churning sampling by *argmax* can 1), alleviate the misalignment between schedulers. 2), boost sampling performance in low-NFE.** First, we visualize the progressive chain of changes when sampling with a scheduler $\kappa_t$ that differs from the linear scheduler used during training. Our sampling process uses 50 steps and a CFG scale of 3. Second, we demonstrate that applying the `argmax` operation to logits can significantly reduce the occurrence of remaining `[MASK]` tokens after sampling.

