# OpenReview forum: "Discrete Interpolants: Unifying the Masked Generative and Discrete Diffusion Models"
_TMLR — Rejected by TMLR_

### Review · Reviewer_atTy · 2025-12-06

**Summary Of Contributions:**

This paper proposes a new framework called “Discrete Interpolants” that tries to connect two popular families of generative models: masked generative models and discrete diffusion models. The authors analyze the design space, such as noise schedule, timestep modeling, temperature, and sampling strategy. They also show that their framework can move from explicit timestep models to implicit timestep models. In addition, they treat image segmentation as an unmasking process, which allows training a joint model for both generation and segmentation. Finally, the evaluation results show that their proposal can achieve state-of-the-art or competitive results on several image and video datasets.

Strengths:
+ Unifying two important generative model families is interesting and the proposed framework is flexible
+ Strong empirical results on large scale datasets

Weaknesses:
- The paper is very theory-heavy, and might be hard for broader audience without strong background to follow
- The practical benefits compared to standard diffusion or masked models are not very clear

**Additional Comments:**

The paper is well-motivated and covers many design choices. While the theory is hard for me to fully verify, the application to segmentation and the strong results make it a good contribution.

**Audience:**

Yes

**Audience Explanation:**

Generative models are a very hot topic. Researchers working on Vision Transformers, Diffusion Models, and multi-modal learning will find this unification work very interesting.

**Broader Impact Concerns:**

I don’t see major ethical concerns.

**Claims And Evidence:**

Yes

**Claims Explanation:**

The authors provide extensive experiments. They compare their methods against strong baseline on standard benchmarks. They report quantitative metrics (FID scores, etc.) which support their claim of achieving state-of-the-art performance. They also include ablation studies to show how different parts of their design work.

**Requested Changes:**

The authors could include the efficiency discussions. Considering the practicality in real-world scenarios, the paper could be strengthened by adding the comparison of training cost (GPU hours) and inference speed (latency/throughput) to the baselines. For example, does the “implicit timestep” design make inference slower or faster?

---

### Review · Reviewer_k6SN · 2025-12-29

**Summary Of Contributions:**

## Summary

The paper introduces **Discrete Interpolants**, a framework designed to bridge the gap between two prominent generative paradigms in the vision domain: **Masked Generative Models (MGM)** and **Non-Autoregressive (Discrete Diffusion) Models**. While previous work on discrete flow matching primarily focused on language or small-scale datasets, this research scales the approach to large-scale vision datasets like **CC12M** and **ImageNet256**.

## Strengths

* **Scaling**: This paper provides successful scaling of discrete flow matching on large-scale datasets, unlike prior works that stopped at CIFAR-10.
* **Theoretical Unification**: Discrete-state modeling is presented as the conceptual bridge between diffusion and MGM. The introduction of Implicit Timestep Models is a useful insight, as it explains why timestep dependence can be removed in discrete settings since the masking ratio inherently carries information about the noise level (or timestep).
* **Empirical Analysis**: The authors conduct thorough ablation studies on various design choices, including noise schedules, softmax temperature, and Classifier-Free Guidance (CFG).

## Weaknesses

* **Inherited Limitations of Mask-Based Methods**: As the authors acknowledge [*in B Potential Impact*] , a common drawback of mask-based approaches is that unmasked tokens cannot be "masked back", meaning errors made during early sampling steps can accumulate without a mechanism for correction.
* **Efficiency Gap in Sampling**: Although the paper explored MGM-style sampling to reduce the number of function evaluations (NFE), a significant disparity in efficiency remains. Standard discrete flow matching often requires thousands of steps, whereas heuristic MGM methods like MaskGiT achieve results in far fewer, highlighting a need for further optimization in the diffusion-based sampling process.
* **Complexity of Schedule Selection**: While the paper explores many masking schedules (linear, cosine, quadratic etc.), it notes [*in Ablate Sampling Scheduler on Page 9*] that misalignment between training and sampling schedules leads to decreased performance. This suggests that the framework is sensitive to these choices, requiring careful tuning for different datasets and tasks.

**Audience:**

Yes

**Audience Explanation:**

* TMLR’s audience would be interested in this work because it provides a unifying theoretical framework that bridges Masked Generative Models (MGM) and discrete-state diffusion models, paradigms that are currently central to the generative modelling community,. The paper’s contribution to scalability in the vision domain is particularly significant.
* Additionally, the shift from Explicit to Implicit Timestep Models demonstrates that timestep dependence can be removed in discrete settings, offering practical advantages for cleaner feature extraction and more flexible, task-specific sampling.
* Finally, the framework’s ability to unify generative and discriminative tasks by reframing segmentation as an unmasking process suggests a versatile architecture capable of handling multi-modal joint distributions with a single training stage.

**Broader Impact Concerns:**

No concerns.

**Claims And Evidence:**

Yes

**Claims Explanation:**

As also briefly stated in the "Strength" above, the primary claims are supported as follows:
* **Experiments on Large Datasets**: The authors provide clear evidence on MS COCO (achieving a competitive FID of 5.31) and CC12M (12 million images), where their implicit timestep model outperforms baselines like MDM and Simple Diffusion.
* **Methodological Refinements**: The authors conduct extensive ablation studies on design choices such as Classifier-Free Guidance (CFG), softmax temperature, and masking schedules.
* **Cross-Modality Versatility**: Beyond still images, the authors provide evidence of the framework's efficacy in video generation using the FaceForensics and DMLab datasets, outperforming continuous-state counterparts like Latte in video distance metrics.
* **Task Unification**: The paper demonstrates that image segmentation can be reframed as an unmasking process. On the Cityscapes dataset, their framework achieves a high mIOU of 90.1, showing that a single model trained on joint distributions can handle both generative (mask-to-image) and discriminative (image-to-mask) tasks.

**Requested Changes:**

## Critical to Securing Recommendation for Acceptance

1. **Clarification of the Efficiency Gap**: The authors acknowledge in the appendix a "substantial disparity in sampling efficiency," noting that Discrete Flow Matching often requires 2,048 steps compared to MaskGiT’s 18. Given that the paper aims to unify these paradigms, it is critical that the main text provides a more prominent discussion on this gap and whether the proposed Implicit Timestep Model (ITM) or the argmax churning technique can realistically bridge this gap in practical applications without sacrificing FID.

2. **Integration of the "No Masking Back" Solution**: The authors note a key limitation: once a token is unmasked, it cannot be masked back, which can lead to accumulating denoising errors. While they suggest a "smoothing factor" in the appendix to address this, this theoretical extension should be more clearly integrated into the main framework.

## Strengthening the Work (Suggested Adjustments)

3. **Empirical Validation of "Cleaner Features"**: One of the claimed advantages of the Implicit Timestep Model is that features from such models are "*cleaner and more suitable for downstream discriminative tasks*". The paper would be significantly strengthened if the authors included a small experiment comparing the performance of ITM features versus ETM features on a standard downstream task (e.g., linear probing on ImageNet) to prove this point.

4. **Comparative Analysis with Hybrid Models**: The appendix mentions recent works like Show-o and Transfusion, which also attempt to unify multimodal tasks. Moving a brief comparison of these models into the main "Related Work" section would better situate Discrete Interpolants within the current landscape of single-stage joint training models.

5. **Sensitivity of Scheduler Misalignment**: The authors show that shifting the sampling scheduler from the one used in training (e.g., linear to cosine) decreases performance, though argmax helps. A more detailed table or graph showing the performance decay across all combinations of training/sampling schedules would provide valuable practitioners' insights.

---

### Review · Reviewer_2hPz · 2026-02-06

**Summary Of Contributions:**

The paper proposes “Discrete Interpolants,” a framework for discrete-state generative modeling that aims to place masked generative models (MaskGIT-style iterative unmasking) and discrete diffusion/flow-matching methods within a shared design space. It defines a discrete corruption path between a masked prior and the data tokens using a schedule, trains a model to predict clean tokens from partially corrupted inputs with cross-entropy (including variants with masking-only and time weighting), and studies two parameterizations: an explicit timestep-conditional predictor and an implicit predictor that drops the timestep dependency. The authors also experiment with a masked generative modeling-style sampling. The paper evaluates this family of models on standard vision benchmarks (COCO, ImageNet), a large-scale text-to-image setting (CC12M), and also explores a joint image–segmentation-token formulation to perform image segmentation, accompanied by ablations over sampling steps, temperature, guidance strength, and schedules.

**Audience:**

Yes

**Audience Explanation:**

Researchers and practitioners working on discrete generative modeling for vision, especially those interested in the relationship between masked generative models and discrete diffusion/flow-matching approaches, would likely be interested in the empirical comparisons and ablations. However, in its current form the paper’s impact is limited by unclear writing, ill-specified technical presentation, and some concerns in the experimental setup, so the findings would need clearer formulation and more controlled evaluation to be broadly informative.

**Broader Impact Concerns:**

The paper presents a discrete generative modeling approach primarily for image and video generation. While generative AI systems can be used for harmful purposes, there are no specific concerns or ethical implications of this work that need to be highlighted.

**Claims And Evidence:**

No

**Claims Explanation:**

The authors have clearly put a lot of work in running experiments on different tasks and datasets, but several core claims (novelty, “unifying design space,” and the soundness of the proposed framework) are not convincingly supported because the presentation is currently hard to verify and some comparisons are not clearly presented.

### The writing in this paper is poor and difficult to follow

The paper has many grammatical errors and many sentences do not follow a coherent flow or logical progression, which makes it difficult to track definitions, assumptions, and what is claimed to be new. I provide some examples below (vague claims, unclear referents, and abrupt transitions), but these issues are present throughout most of the paper.

 - The introduction section is difficult to follow and currently does not motivate the work clearly. For example, the sentence about “training and sampling similarity between Diffusion Models and Masked Generative Models” is asserted without specifying which similarities are meant (objective form, corruption process, sampling heuristics, etc.) or providing concrete citations and explanation. The subsequent claim that there is lack of analysis of the shared design space and theoretical underpinnings is also not very clear - which specific factors does this paper analyze? How does scaling the experiments to large vision datasets help provide new insights?

- The related works section does not provide a good explanation of existing work. For example, the first paragraph which is supposed to discuss discrete diffusion models does not provide any meaningful discussion. Why are works that connect diffusion models and autoregressive models relevant? The paper mentions multiple times here that they provide a unified design space for diffusion models and masked generative models without much explanation. The next paragraph supposedly discusses this connection but again does not provide concrete information.

### Section 3: presentation and mathematical issues undermine the central technical claims

- **Type/state-space definitions are currently incorrect or misleading.** The paper describes discrete token data but places it in $\mathbb{R}^L$. For discrete sequences this should be $[K]^L$ (vocabulary size $K$) or a one-hot representation in $\{0,1\}^{K\times L}$. This is important since affects whether later quantities like $\delta_{x_1}$, $p_{1|t}$, and the vector field are well-defined.
- **The “single-token” exposition is not consistent with the sequence-level notation used in key equations.** The paper says it mainly considers the single token setting, yet writes objectives and conditionals such as $p_{1|t}(x_1 | x_t,t;\theta)$ where $x_1$ is introduced earlier as a length-$L$ vector. The factorized multi-token presentation in the Appendix helps, but the presentation in the main text seems incorrect/under-specified.
- **The vector field is written in a way that reads like it decodes $x_t$ to $x_1$ directly, which is confusing/incorrect for DFM.** The paper defines a vector field as $u_t(x_t)$. In discrete flow matching, the corresponding object is the velocity field which is a *per-coordinate* rate function over candidate token values. As written, the specific token that is decoded is not specified.
- **The loss definition is presented inconsistently and likely contains a sign error.** I believe Eq. (2) has a typing error where the minus sign is omitted. The multi-token loss in Eq. (10) is missing the specific token $x^i$ where $i$ denotes the token index. It is possible the authors are using a different notating than the discrete flow matching paper, in that case it should be clearly stated to avoid confusion.

### Novelty and attribution: currently not supported

- **The claimed contributions do not match the contents of the paper.** The name discrete interpolants seems to imply mapping from arbitrary prior distribution to some target distribution (like the stochastic interpolants framework [2]), but this framework uses the fixed absorbing state (all masked tokens) as the prior. The contribution of this work seems to be adapting design choices from masked generative modeling to improve discrete diffusion, rather than providing a theoretical connection between these two approaches or a “interpolants” framework for discrete models.
- **Section 3 reads largely as a restatement of Discrete Flow Matching [1], but with weaker notation and incomplete credit.** The paper explicitly notes it builds on discrete flow matching, but the section does not clearly isolate what is genuinely new versus what is adapted from prior work. This is particularly important because the method is introduced as a new framework called Discrete Interpolants.
- **Masking and weighting are presented as modifications, but they are not new in themselves and should be positioned as prior art.** Masked-token cross-entropy training is central to MaskGIT-style models [3]. Likewise, loss/timestep weighting strategies have a long history in diffusion training (e.g., importance sampling / weighting schemes in Improved DDPM [4] and related work such as Min-SNR weighting [5]. If the novelty is in a specific combination or in discrete settings, that needs to be stated precisely.
- **The implicit-vs-explicit timestep discussion is framed as an advantage/property of the proposed method, but closely related observations exist in prior work.** The paper argues that the timestep dependency can be removed with similar optima, and it also acknowledges related prior work. This section needs a clearer statement of what is new beyond prior analyses and prior connections to MaskGIT-style sampling (e.g., simplified sampling discussions have appeared in related work [6]).

### Experimental evidence: not yet sufficient to support strong comparative claims

- **Key comparisons appear unfair due to different sampling compute (NFEs).** The training details indicate sampling steps of “ITM/ETM 20/1k.” Yet Table 2 compares against baselines such as U-ViT from Bao et al. 2023 which uses 50 steps. Since image quality/FID typically improves substantially with more steps, comparisons should be made at matched NFE or at least presented with compute-quality tradeoffs. Without this, it’s hard to attribute gains to the method rather than to higher sampling budgets.
- **It is not always clear which sampling procedure is used for which result.** The paper describes both diffusion-like sampling and MGM-style masking sampling, and Figure 2 explicitly compares ETM/ITM to MGM-style variants. However, the main tables do not clearly specify whether reported results use ETM, ITM, MGM-style, or a hybrid.
- **Ablations largely sweep known parameters without providing new insights.** Varying NFE, temperature, CFG, schedule mismatch, etc. is useful engineering information, but these are widely explored in both diffusion and masked modeling communities. The paper should either articulate a genuinely new insight that emerges from the unified framing or present the ablations more explicitly as a systems/practitioner guide.
- **Tokenizer confound is not addressed.** The method uses the SD-VQ-F8 / SD-VQGAN family tokenizer trained at a large scale. It is not clear if the baselines may not use the same tokenizer. Without a controlled study, improvements can plausibly be explained by the tokenizer choice rather than the proposed framework.
- **Baselines and evaluation context could be strengthened.** Some comparisons appear to rely on older baselines, and the paper should include stronger recent baselines. For instance, more recent AR-based generation work can be quite strong on ImageNet (e.g., RAR [7])

*[1] Gat, Itai, et al. "Discrete flow matching." Advances in Neural Information Processing Systems 37 (2024): 133345-133385.*

*[2] Albergo, Michael S., Nicholas M. Boffi, and Eric Vanden-Eijnden. "Stochastic interpolants: A unifying framework for flows and diffusions." arXiv preprint arXiv:2303.08797 (2023).*

*[3] Chang, Huiwen, et al. "Maskgit: Masked generative image transformer." Proceedings of the IEEE/CVF conference on computer vision and pattern recognition. 2022.*

*[4] Nichol, Alexander Quinn, and Prafulla Dhariwal. "Improved denoising diffusion probabilistic models." International conference on machine learning. PMLR, 2021.*

*[5] Hang, Tiankai, et al. "Efficient diffusion training via min-snr weighting strategy." Proceedings of the IEEE/CVF international conference on computer vision. 2023.*

*[6] Shi, Jiaxin, et al. "Simplified and generalized masked diffusion for discrete data." Advances in neural information processing systems 37 (2024): 103131-103167.*

*[7] Yu, Qihang, et al. "Randomized autoregressive visual generation." Proceedings of the IEEE/CVF International Conference on Computer Vision. 2025.*

**Requested Changes:**

The paper in its current form requires major revision for me to recommend acceptance. The most critical changes are discussed below.

- Rewrite and proofread the paper for grammatical correctness and clearer logical flow, especially in the abstract, introduction, and related work sections. I suggest the authors make sure that definitions should precede usage, variables and notation remains consistent, and claims are made with relevant explanation and citations.
- Rewrite Section 3 to be mathematically correct and consistent: define the discrete state space correctly, use sequence-level notation to avoid confusion, explicitly define terms and quantities with correct arguments and fix the training loss equation. Sections 3.1 and parts of 3.2 that largely discuss discrete flow matching can also be introduced in a separate “Background” section if needed.
- Rename the method or make it clear why the method falls within the “interpolants” framework, since it does not bridge two arbitrary distributions. The contribution of this work seems to be adapting certain design choices from masked generative models to improve discrete diffusion. If so, then the introduction and contributions need major revision to align with the contents of the paper.
- Improve attribution and novelty positioning throughout the paper to clearly separate what is adapted from discrete flow matching / masked generative modeling and what is new, credit DFM at the point of derivation, and avoid presenting masking, loss weighting, or implicit time dependence as novel unless the paper demonstrates what materially differs from prior work (please see comments above for specific details).
- Make the experimental comparisons clear and reproducible by specifying exactly which sampler (ETM/ITM/MGM-style) is used for each table, add matched-NFE results or compute–quality tradeoff curves (given that ETM uses up to 1k steps in the provided details). In addition Figure 2 seems to have much higher FID on Imagenet than the results presented in Table 3, please add an explanation or fix as needed.
- Disentangle confounding factors such as tokenizer, preferably by using the same tokenizer for all methods, and add stronger/more recent baselines where applicable (e.g., modern AR or masked-model baselines beyond older 2023–2024 comparisons).

---

### Decision · Action_Editor_5QWh · 2026-04-08

**Recommendation:** Reject

**Additional Comments:**

The recommendation for rejection is based on the following:

- Lack of Author Engagement. The authors didn't provide a rebuttal or update the manuscript to address the reviewers concerns. The lack of revision in response to critical technical flaws makes the paper unsuitable for publication.
- Unresolved Technical Issues. Critical issues with notation and definitions identified by the reviewers remain in the text, making the methodology hard to verify.
- Clarity. The paper suffers from poor writing and fails to adequately credit or differentiate itself from prior art, leading to concerns regarding the significance of its contribution.

**Audience:**

Yes

**Audience Explanation:**

The topic of unifying Masked Generative Models and discrete-state diffusion models is of high interest to the generative modeling community. Researchers working on vision transformers and large-scale discrete generative models would find the scaling analysis and the concept of "Implicit Timestep Models" relevant. However, the interest is currently overshadowed by the lack of technical reliability and clarity in the present manuscript.

**Claims And Evidence:**

No

**Claims Explanation:**

While the paper presents experimental results on large-scale datasets, several critical technical and methodological issues undermine the validity of the evidence:
- Mathematical Inaccuracies. As highlighted by Reviewer 2hPz, there are fundamental errors in the technical presentation, such as incorrectly defining discrete state spaces and inconsistencies in the training loss equations (e.g., potential sign errors and missing indices).
- Methodological Flaws. The experimental comparisons appear to be poorly controlled. Specifically, the proposed method uses a significantly higher sampling budget (Number of Function Evaluations) compared to the baselines, making it impossible to determine if the performance gains are due to the framework or simply increased computation.
- Unclear Contribution. The manuscript fails to sufficiently distinguish its "Discrete Interpolants" framework from existing work, specifically Discrete Flow Matching (DFM). Without a clear delta or more rigorous theoretical grounding, the claim of a new unifying framework is not supported.

**Resubmission Of Major Revision:**

The authors may consider submitting a major revision at a later time.